# Netrin-1 regulates the balance of synaptic glutamate signaling in the adult ventral tegmental area

**Marcella M Cline[1,2], Barbara Juarez[1,3], Avery Hunker[3†], Ernesto G Regiarto[3], Bryan Hariadi[3], Marta E Soden[3], Larry S Zweifel[1,3]***

[1]Department of Psychiatry and Behavioral Sciences, University of Washington, Seattle, United States; [2]Molecular and Cellular Biology Program, University of Washington, Seattle, United States; [3]Department of Pharmacology, University of Washington, Seattle, United States

**Abstract** The axonal guidance cue netrin-1 serves a critical role in neural circuit development by promoting growth cone motility, axonal branching, and synaptogenesis. Within the adult mouse brain, expression of the gene encoding (*Ntn1*) is highly enriched in the ventral midbrain where it is expressed in both GABAergic and dopaminergic neurons, but its function in these cell types in the adult system remains largely unknown. To address this, we performed viral-mediated, cell-type specific CRISPR-Cas9 mutagenesis of *Ntn1* in the ventral tegmental area (VTA) of adult mice. *Ntn1* loss-of-function in either cell type resulted in a significant reduction in excitatory postsynaptic connectivity. In dopamine neurons, the reduced excitatory tone had a minimal phenotypic behavioral outcome; however, reduced glutamatergic tone on VTA GABA neurons induced behaviors associated with a hyperdopaminergic phenotype. Simultaneous loss of *Ntn1* function in both cell types largely rescued the phenotype observed in the GABA-only mutagenesis. These findings demonstrate an important role for Ntn1 in maintaining excitatory connectivity in the adult midbrain and that a balance in this connectivity within two of the major cell types of the VTA is critical for the proper functioning of the mesolimbic system.

**\*For correspondence:**
larryz@u.washington.edu

**Present address:** †Allen Brain Institute, Seattle, United States

**Competing interest:** The authors declare that no competing interests exist.

## Editor's evaluation

This manuscript reports an important, previously unappreciated, non-developmental role for the guidance cue netrin-1 in midbrain physiology and related behavior in adult animals. Using multiple experimental tools in adult mice, the study convincingly shows that netrin-1 within midbrain dopamine and GABA neurons is necessary to maintain dopamine excitatory tone and plays a role in motivated and anxiety-like behavior. This paper will be of interest to neuroscientists studying dopamine function and/or motivated behavior and those interested in ways that neurodevelopmental genes can continue to play a role in neuronal function and behavior into adulthood.

## Introduction

Proper regulation of the midbrain dopamine system is essential for numerous brain functions and behavior (*Bissonette and Roesch, 2016*). Disruption in the balance of midbrain dopamine neuron activity has been linked to several neurological and psychiatric conditions, including autism (*Pavăl, 2017*), schizophrenia (*Hietala and Syvälahti, 1996*), and substance use disorders (*Ostroumov and Dani, 2018*). Within the VTA, the activity of dopamine neurons is regulated in part by inhibitory (GABAergic) and excitatory (glutamatergic) synaptic input. The molecular mechanisms that maintain

the balance of inhibitory and excitatory connectivity in the adult midbrain, however, remain poorly resolved.

Genome-wide association studies and analysis of de novo mutations have strongly implicated genes regulating neuronal axon guidance in neurodevelopmental disorders (*Gilman et al., 2012*; *Gulsuner et al., 2013*). Although the impact of mutations in these genes early in development is likely critical for their role in neurodevelopmental disorders, many of the genes maintain high levels of expression in the adult brain, and their functions in this context are less understood. We previously demonstrated that the axonal guidance receptor Robo2 is necessary for the maintenance of inhibitory synaptic connectivity in the adult VTA (*Gore et al., 2017*), suggesting that axonal guidance proteins have a critical function in maintaining synaptic connectivity in the adult midbrain.

Netrin-1 is predominantly recognized for its role in neurodevelopmental processes (*Gore et al., 2017*; *Manitt et al., 2010*; *Winberg et al., 1998*; *Glasgow et al., 2018*; *Yetnikoff et al., 2010*). During development, the gene encoding netrin-1 (*Ntn1*) is highly expressed throughout the central nervous system (CNS). Following this critical period global expression decreases (*Manitt et al., 2010*), but expression within the limbic system, particularly in the ventral midbrain, persists. Consistent with the continued function of Ntn1 following early development, genetic inactivation of either *Ntn1* (*Winberg et al., 1998*) or its receptor *Dcc Glasgow et al., 2018* from forebrain glutamatergic neurons in late postnatal development results in significantly impaired spatial memory in adult mice that corresponds to a loss of hippocampal plasticity. Within the VTA, *Dcc* expression levels in adult mice are significantly upregulated following amphetamine exposure (*Yetnikoff et al., 2010*), and *Dcc* haploinsufficient mice display blunted locomotor response to amphetamine (*Flores et al., 2005*), consistent with increased excitatory synaptic strength in the VTA following amphetamine treatment (*Saal et al., 2003*). These results suggest a potential role for Ntn1 signaling through Dcc in regulating excitatory tone in the adult dopamine system.

To determine whether Ntn1 regulates excitatory synaptic connectivity in the VTA of adult mice, we used viral-mediated, Cre-inducible CRISPR/Cas9 (*Hunker et al., 2020*) to selectively mutate *Ntn1* in midbrain dopamine and GABA neurons. We find that *Ntn1* loss of function significantly reduces postsynaptic glutamate receptor-mediated currents in a cell-autonomous manner similar to what has been reported previously in the adult hippocampus (*Glasgow et al., 2018*). We further show that *Ntn1* loss of function in VTA GABA neurons has a more profound effect on behavior than the loss of function in VTA dopamine neurons. Intriguingly, the simultaneous loss of function of *Ntn1* in both cell types of the VTA largely rescues the behavioral phenotypes observed following mutagenesis in VTA GABA neurons alone. These data support a model in which the balance of excitatory synaptic connectivity between dopamine and GABA neurons within the VTA is maintained by the persistent expression of the developmental gene *Ntn1*. This continued function of *Ntn1* in adulthood sustains the excitatory/inhibitory equilibrium onto dopamine neurons that is critical to the function of the mesolimbic dopamine system.

## Results

### *Ntn1* expression and mutagenesis in the VTA

In situ hybridization analysis of *Ntn1* from the Allen Institute mouse brain expression atlas (*Lein et al., 2007*) shows diffuse and low levels of expression throughout the adult mouse brain, with moderate expression levels in the cerebellum and hippocampus (*Figure 1A*), and the highest level of expression in the ventral midbrain (substantia nigra and ventral tegmental area). The VTA is comprised of multiple cell types *Morales and Margolis, 2017*; to determine the cell type-specific expression of *Ntn1* within the heterogeneous VTA, we performed RNAscope in situ hybridization on midbrain slices from adult wild-type mice (>8 weeks of age) and probed for *Ntn1, Th* (tyrosine hydroxylase, a marker of dopamine neurons), and *Slc32a1* (vesicular GABA transporter [Vgat], a marker of GABA neurons). We found *Ntn1* expression to be present throughout the VTA, largely localized to *Th*-positive neurons but also present in GABA neurons (*Figure 1C–F*). Of the identified Ntn1 positive cells, *Ntn1* expression co-localized with *Th* expression (dopamine producing neurons; 72.2% co-localization) and *Slc32a1*-expressing GABA neurons (18.1% co-localization) (*Figure 1E*). The remaining *Ntn1* expressing cells that do not co-localize with *Th* or *Slc32a1* are likely glutamatergic neurons

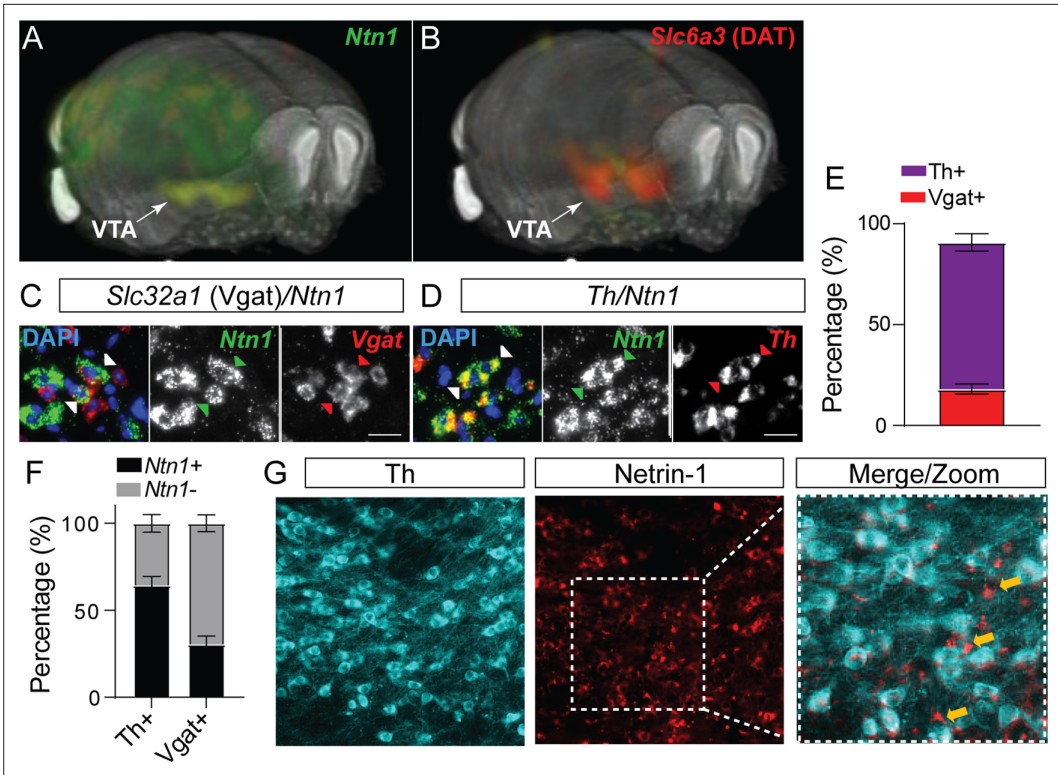

**Figure 1.** Netrin-1 is present in the adult ventral tegmental area (VTA) and expressed by both dopamine and GABA neurons. 3D display of *Ntn1* (**A**) and *Slc6a3* (B, dopamine marker) from the Allen Brain Atlas. (**C–D**) 20 X magnification images of in situ hybridization (RNAScope) for *Ntn1* (green) and *Slc32a1* (GABA marker; red, **C**) and *Th* (dopamine marker; red; **D**). Arrows indicate co-labeling of *Ntn1* with *Slc32a1* (**C**) or *Th* (**D**). Scale bar indicates 20 µm. (**E–F**) Quantification of cell type expression. Of the cells expressing *Ntn1*, 72.2% were dopaminergic (*Th+*) and 18.1% were GABAergic (*Slc32a1+*; **E**). (**F**) Of the total of *Th+* identified cells, 64.5% co-expressed Ntn1 (35.6% did not express Ntn1), and 30.4% of *Slc32a1* identified cells co-expressed *Ntn1* (69.5% did not express Ntn1). (**G**) Immunohistochemistry confirms the presence of Ntn1 protein (red) in both Th+ (cyan) and non-dopamine cells (Th- cells, indicated by yellow arrows).

The online version of this article includes the following source data for figure 1:

**Source data 1.** Cell counts.

(*Morales and Margolis, 2017*), or possibly glial cells (*Phillips et al., 2022*). Immunohistochemistry for Ntn1 and Th (*Figure 1G*) confirmed the presence of Ntn1 in dopamine and non-dopamine producing (Th-negative) cells.

To selectively mutate *Ntn1* in specific cell types in the VTA, we designed a single guide RNA (sgRNA) targeting exon 2 in mice (sg*Ntn1*; *Figure 2A*) and cloned it into an AAV packaging plasmid containing a Cre-recombinase dependent expression cassette for SaCas9 (*Hunker et al., 2020*). To determine the efficiency of *Ntn1* mutagenesis, we injected DAT-Cre (*Slc6a3*[Cre/+]) mice (aged 8–10 weeks) bilaterally into the VTA with either AAV-FLEX-SaCas9-HA-sg*Ntn*1 and AAV-FLEX-YFP (DAT-Cre *Ntn1*-cKO mice) or AAV-FLEX-SaCas9-sg*Rosa26* (a gene locus with no known function; control mice). Four to five weeks following injection, we performed immunohistochemistry for Ntn1 and Th. *Ntn1* conditional knockout (cKO) resulted in a significant reduction in the proportion of VTA Th-positive cells co-labeled with Ntn1 in DAT-Cre *Ntn1* cKO mice compared to controls (*Figure 2D–E*). In contrast to previous findings following *Ntn1* deletion in the substantia nigra (*Jasmin et al., 2021*), the average number of Th + cells per slice was not statistically different in DAT-Cre *Ntn1* cKO mice compared to controls (control: 178.2 ± 12.68 and *Ntn1* cKO 173.3 ± 10.75). Although, this result is consistent with *Ntn1* inactivation not compromising cell viability, without a complete stereological analysis of every neuron within the VTA, we cannot definitively conclude that some cell loss did not occur.

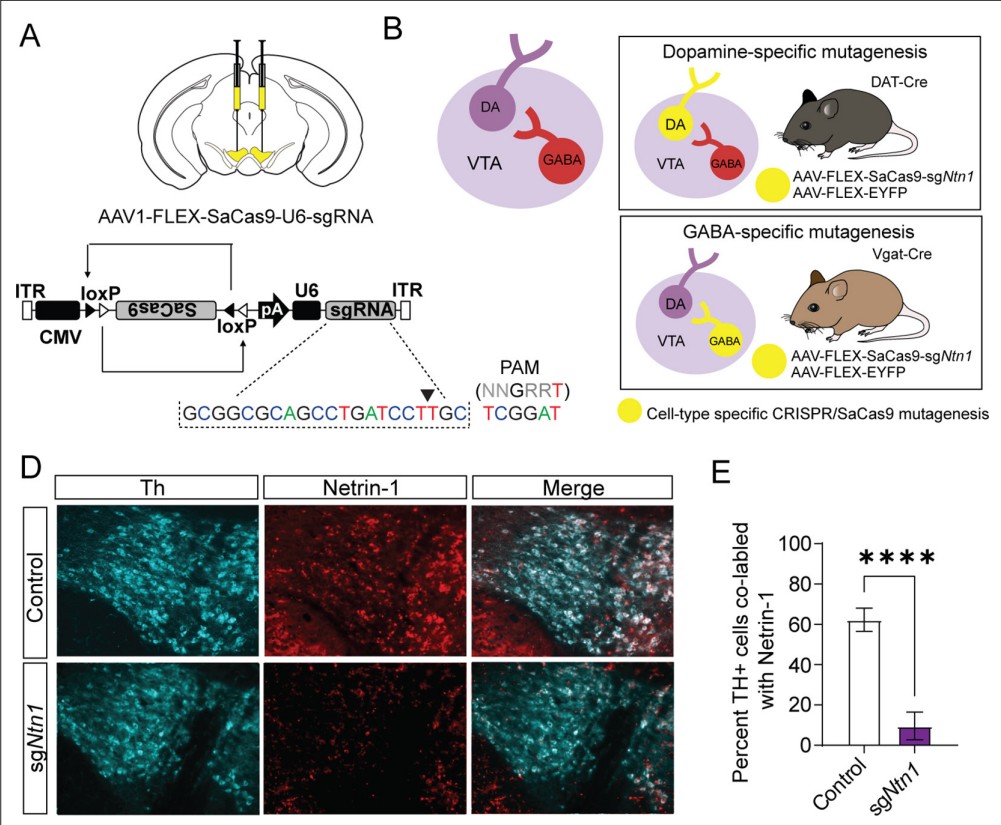

**Figure 2.** Virally delivered CRISPR-Cas9 complex targeting the *Ntn1* locus results in a significant reduction in Ntn1 antibody staining. (**A–B**) Schematics summarizing cell type-specific knockout procedure. (**A**) Adult mice were injected bilaterally into the VTA with AAV-FLEX-SaCas9-HA-sg*Ntn1* and AAV-FLEX-YFP. Control mice received an equivalent volume of -sgRosa26 and/or AAV-FLEX-YFP. SaCas9 is virally delivered into the genome in the inactive orientation and returned to the active orientation only in the presence of Cre recombinase, limiting Cas9 expression to target cells. (**B**) Schematic of the VTA (left) showing VTA GABA neurons project to and inhibit VTA dopamine neurons. By using transgenic Cre-driver mouse lines (right) viral delivery of SaCas9 results in gene disruption in specifically VTA dopamine neurons (DAT-Cre mice, top panel), or VTA GABA neurons (Vgat-Cre mice, bottom panel). (**D**) Example images for Th (cyan) and Ntn1 (red) immunostaining in the ventral tegmental area (VTA) of mice injected with control or sgNtn1 CRISPR virus. (**E**) Quantification of the percentage of Th + cells co-labled with Ntn1 (Students *t*-test; t=8.179, df = 10, 62.25 ± 5.796 vs 9.586 ± 2.807, ****p<0.0001).

The online version of this article includes the following source data for figure 2:

**Source data 1.** Cell counts.

## Netrin-1 regulates excitatory connectivity within the adult VTA

Previous research has shown that Ntn1 regulates excitatory synaptic connectivity in the adult hippocampus (*Glasgow et al., 2018*). To determine the impact of *Ntn1* loss of function on synaptic connectivity, DAT-Cre or Vgat-Cre (*Slc32a1*$^{Cre/+}$) mice were injected with AAV1-FLEX-SaCas9-U6-sg*Ntn1* and AAV1-FLEX–YFP (*Figure 3A and E*). After at least four weeks, miniature excitatory postsynaptic currents (mEPSCs) were recorded from fluorescently identified dopamine or GABA neurons of the VTA. *Ntn1* mutagenesis in dopamine neurons resulted in significantly reduced mEPSC amplitude and frequency (*Figure 3B–D*). Similarly, *Ntn1* mutagenesis in VTA GABA neurons also resulted in significantly reduced mEPSC amplitude and frequency (*Figure 3F–H*). We did not detect significant effects on miniature inhibitory postsynaptic currents (mIPSCs) in VTA dopamine or GABA neurons following *Ntn1* mutagenesis in these cells (*Figure 3—figure supplement 1*), suggesting Ntn1 does not play a role in regulating inhibitory connectivity in these cells.

Because Ntn1 is a secreted protein, it is also possible that *Ntn1* loss of function in one cell type could affect synaptic connectivity in adjacent neurons in which the gene was not inactivated, inducing a non-cell autonomous effect. To address this, we recorded mEPSCs from non-YFP-expressing

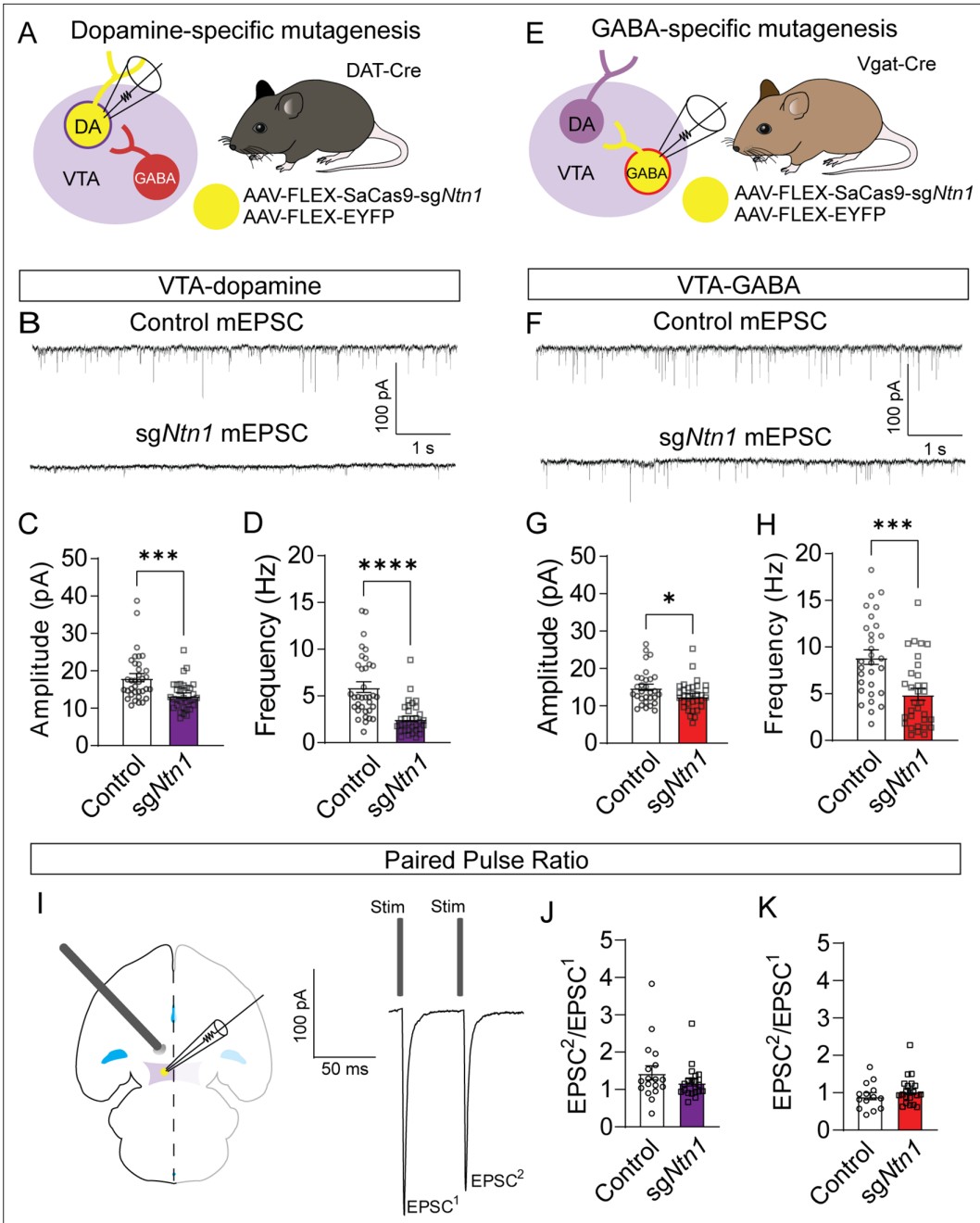

**Figure 3.** Loss of *Ntn1* results in a significant reduction in excitatory postsynaptic current. (**A**) Schematic of DAT-Cre dopamine specific Ntn1cKO. (**B**) Sample traces from control (top panel) and DAT Ntn1 cKO mice (bottom panel). (**C–D**) mEPSC amplitude (**C**) and frequency (**D**) measured from fluorescently identified dopamine neurons (n=35 controls, n=33 cKO, t=3.744, df = 66, ***p<0.001 and t=5.259, df = 66, ****p<0.0001). (**E**) Schematic of Vgat-Cre GABA specific Ntn1cKO. (**F**) Sample traces from control (top panel) and Vgat Ntn1 cKO mice (bottom panel). (**G–H**) mEPSC amplitude (**G**) and frequency (**H**) measured from fluorescently identified GABA neurons (n=30 controls, n=32 cKO, t=2.048, df = 60, *p<0.05, and t=3.966, df = 60, ***p<0.001). (**I**) Schematic of stimulating electrode placement in horizontal midbrain slice and example EPSCs. (**J–K**) Paired pulse ratio in dopamine (J, n=18 controls, n=21 cKO, t=1.271, df = 37, p>0.05), or GABA neurons (K, n=14 controls, n=21 cKO, t=1.105, df = 33, p>0.05).

The online version of this article includes the following source data and figure supplement(s) for figure 3:

**Source data 1.** EPSCs and IPSCs from targeted cells.

**Source data 2.** Additional EPSC and IPSC data from non-targeted cells.

*Figure 3 continued on next page*

*Figure 3 continued*

**Figure supplement 1.** No significant differences in inhibitory synaptic connectivity associated with *Ntn1* loss of function.

**Figure supplement 2.** No significant differences in excitatory or inhibitory synaptic connectivity in non-targeted cell types.

**Figure supplement 3.** Loss of Netrin function results in significant decrease in AMPA response.

(presumptively non-dopamine) neurons in DAT-Cre mice injected with *Ntn1* CRISPR or control virus, and from non-YFP-expressing (presumptively non-GABA) neurons in Vgat-Cre injected mice. We did not observe significant non-cell autonomous effects on mEPSCs from non-targeted cells (*Figure 3—figure supplement 2*). Similarly, we also did not observe non-cell autonomous effects on mIPSCs from non-targeted cells (*Figure 3—figure supplement 2*).

The observed reduction in mEPSC frequency suggests that loss of *Ntn1* function could act presynaptically, potentially through postsynaptic Ntn1 secretion (*Glasgow et al., 2018*). To test potential presynaptic changes in vesicle release probability, we analyzed the paired-pulse ratio (PPR) of electrically evoked EPSCs delivered 50 ms apart. *Ntn1* mutagenesis in either dopamine or GABA neurons did not result in a significant change in PPR compared to controls, suggesting no measurable change in presynaptic release (*Figure 3J–K*). To further resolve this question, we analyzed potential changes in quantal size by performing a $1/CV^2$ analysis of the coefficient of variation in the mEPSC amplitude. We did not detect a statistically significant change in $1/CV^2$ associated with *Ntn1* loss in either dopamine or GABA cells, further suggesting netrin manipulation is altering either the number of or the function of postsynaptic AMPA receptors (*Figure 3—figure supplement 3A, B*).

Our data suggest that the observed changes in mEPSCs are likely a reflection of reduced AMPA-type or NMDA-type glutamate receptor levels in postsynaptic cells. To address this, fluorescently identified dopamine neurons from DAT-Cre *Ntn1* cKO mice held at –60 mV, and AMPA-evoked current was measured following bath application of 1 uM AMPA. For NMDA currents, neurons were held at +40 mV and NMDA-evoked current was measured following bath application of 50 µM NMDA. *Ntn1* mutagenesis in DAT-Cre mice resulted in significantly reduced AMPA-evoked current compared to controls (*Figure 3—figure supplement 3C, D*). In contrast, NMDA-evoked responses were similar between the groups (*Figure 3—figure supplement 3E, F*). These results suggest that Ntn1 regulates AMPA receptor availability in adult VTA.

## *Ntn1* loss of function in VTA-dopamine neurons has little effect on behavior

Dopamine producing neurons of the VTA regulate multiple aspects of locomotor activity, motivated behavior, and psychomotor activation. To determine whether conditional mutagenesis of *Ntn1* in dopamine neurons, and subsequent reduction in excitatory synaptic connectivity impacts these behaviors, we injected DAT-Cre mice with AAV1-FLEX-SaCas9-sg*Ntn1* or AAV1-FLEX-SaCas9-sg*Rosa26* (control) and assayed them in multiple behavioral paradigms. First, we monitored day-night locomotion in control and AAV1-FLEX-SaCas9-sg*Ntn1* injected DAT-Cre mice. No significant differences were detected (*Figure 4B* and *Figure 4—figure supplement 1*).

To determine whether appetitive conditioning behaviors are disrupted by the loss of *Ntn1* function in VTA dopamine neurons, we assayed mice in a simple instrumental conditioning paradigm using a fixed-ratio 1 (FR1) followed by a fixed ratio 5 (FR5) schedule of reinforcement in which one or five lever presses are required to obtain a food reward, respectively. We did not observe significant differences in either of these behavioral tasks (*Figure 4C*). Next, we monitored motivated behavior using a progressive ratio schedule of reinforcement in which the number of lever presses required for reinforcement increases non-arithmetically (1, 2, 4, 7, 13, 19, 25, 34, 43, 52, 61, 73…), and again did not observe significant differences between control and experimental mice (*Figure 4D*). Following PR, we reinstated FR1 responding for three days followed by extinction training, and again did not detect any differences between the two groups (*Figure 4E* and *Figure 4—figure supplement 1*), indicating *Ntn1* loss of function in VTA dopamine neurons did not alter appetitive conditioning behaviors. Although appetitive conditioning was not affected by *Ntn1* loss of function in dopamine neurons, we

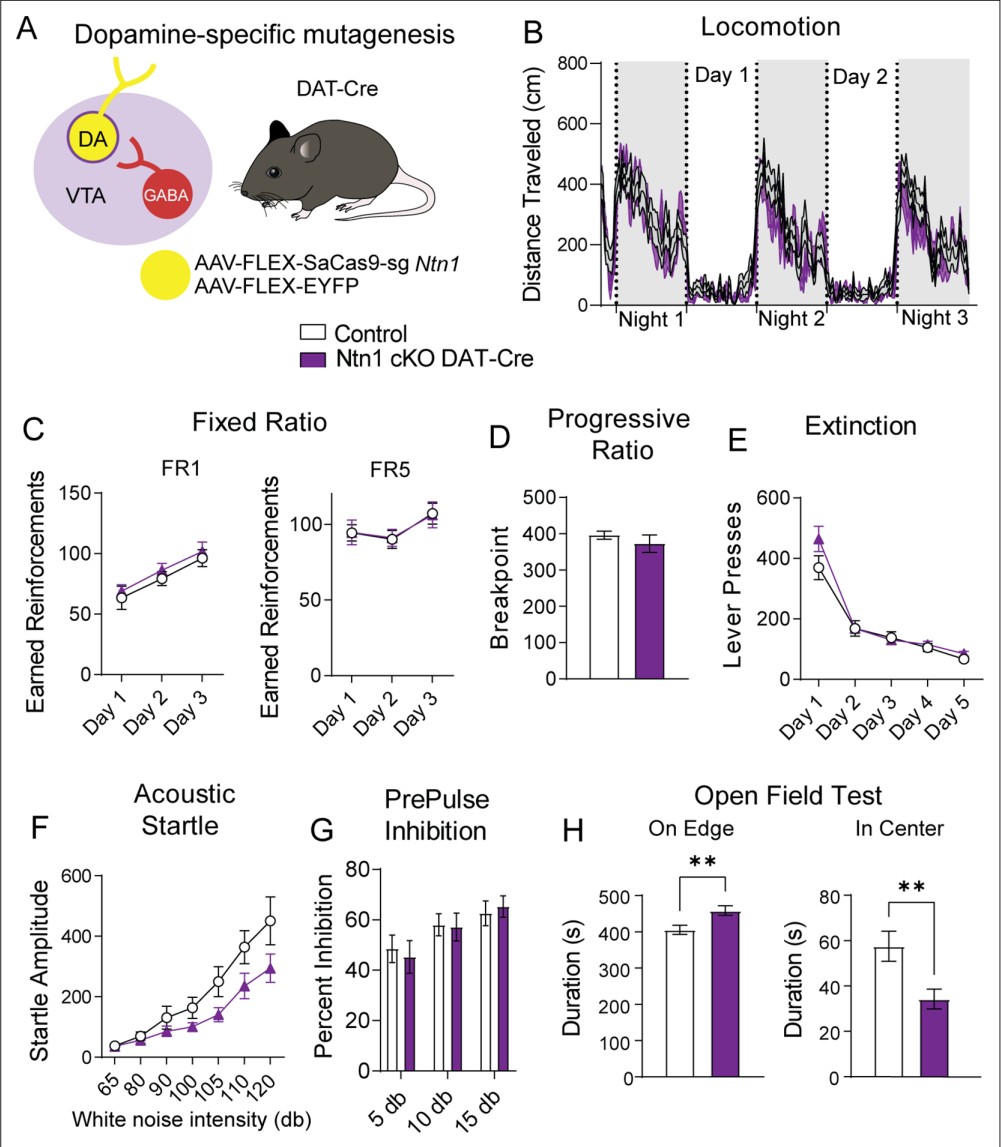

**Figure 4.** *Ntn1* cKO in DA neurons results in little behavioral alteration. (**A**) Schematic summarizing cell type-specific knockout procedure. (**B**) Distance traveled in 15 min bins over the course of three nights and two days (n=21 control; n=15 cKO, Two-way ANOVA, Group F(1, 34)=1.169, p=0.2872, Time F(18.25, 620.6)=21.97 p<0.0001, Interaction F(251, 8534)=1.063 p=0.2380). (**C**) Earned reinforcers during three days of FR1 or FR5 operant conditioning (n=19 control; n=15, FR1; Group F(1, 96)=0.9761 p=0.3257, Time F(2, 96)=9.999 p=0.0001, Interaction F(2, 96)=0.006622 p=0.9934; FR5 Group F(1, 32)=0.6140, p=0.9808, Time F (2, 96)=2.786 p=0.0667, Interaction F(2, 96)=0.008669 p=0.9914). (**D**) Breakpoint (maximum presses per reinforcer) on a progressive ratio task (t=0.9434, df = 32, p=0.3525). (**E**) Lever presses per session during five days of extinction training (Group F(1, 32)=1.336, p=0.2562, Time F(4, 128)=87.55 p<0.0001, Interaction F(4, 128)=2.017 p=0.0959). (**F**) Acoustic startle response to varying intensity white noise stimuli (Group F(1, 31)=3.176 p=0.0845, Intensity F(1.737, 53.83)=37.74 p<0.0001, Interaction F(6, 186)=2.124 p=0.0525) (**G**) Percent inhibition of startle response following pre-pulse at indicated intensities (Group F(1, 96)=0.05032 p=0.8230, Intensity F(2, 96)=5.638 p=0.0048, Interaction F(2, 96)=0.2402 p=0.7870). (**H**) Time on edge or in center of an open field arena during a 10 min test session (Edge: t=2.897, df = 32, **p<0.01, Center: t=2.750, df = 32, **p<0.01).

The online version of this article includes the following source data and figure supplement(s) for figure 4:

**Source data 1.** Behavioral data for *Figure 4*.

**Figure supplement 1.** Additional behavioral analysis of Ntn1 cKO DAT-Cre mice.

**Figure supplement 1—source data 1.** Behavioral data for *Figure 4—figure supplement 1*.

did observe a slight but significant reduction in body weight in these mice relative to controls prior to calorie restriction (*Figure 4—figure supplement 1*).

To determine whether sensory-motor gating is altered in mice with loss of *Ntn1* function in VTA dopamine neurons, we assayed them in acoustic startle and pre-pulse inhibition (PPI) paradigms. Although acoustic startle responses were reduced in AAV1-FLEX-SaCas9-sg*Ntn1* injected mice, this did not reach significance (*Figure 4F*). Moreover, we did not observe differences in PPI percentage inhibition (*Figure 4G*). These results indicate that loss of *Ntn1* function in VTA dopamine neurons does not appear to affect psychomotor activation.

In addition to reinforcement and motivation, dopamine regulates other dimensions of affective behavior. To test whether anxiety-related behavior is affected in experimental mice relative to control mice, we assayed them in an open-field test. AAV1-FLEX-SaCas9-sg*Ntn1* injected DAT-Cre mice spent significantly more time on the edge of the open field arena and significantly less time in the center of the arena, consistent with an elevation in anxiety-like behavior (*Figure 4H*). There were no significant locomotor differences associated with the loss of *Ntn1* function in the open field arena (*Figure 4—figure supplement 1*).

### *Ntn1* loss of function in VTA-GABA neurons affects multiple behaviors

To determine whether reducing excitatory synaptic connectivity onto VTA GABA neurons through the loss of *Ntn1* function in these cells impacts behavior, we injected Vgat-Cre mice with AAV1-FLEX-SaCas9-sg*Ntn1* or AAV1-FLEX-SaCas9-sg*Rosa26* (control) into the VTA as described previously and tested these mice using the same behavioral paradigms described above. In contrast to *Ntn1* mutagenesis in dopamine neurons, this manipulation in VTA GABA neurons resulted in a significant increase in locomotor activity (*Figure 5B* and *Figure 5—figure supplement 1*).

In the FR1 schedule of reinforcement, we did not observe a significant difference between the groups; however, we observed an increase in the number of earned reinforcements in the FR5 schedule in mice with *Ntn1* loss of function in VTA GABA neurons (*Figure 5C*). We also observed an increase in the PR schedule of reinforcement in these mice relative to controls (*Figure 5D*). In contrast to DAT-Cre *Ntn1* cKO mice, pre-calorie restriction body weights in Vgat-Cre *Ntn1* cKO mice did not differ from controls (*Figure 5—figure supplement 1*).

Reinstatement of FR1 responding in Vgat-Cre *Ntn1* cKO following PR was not different than controls (*Figure 5—figure supplement 1*). However, during extinction training, Vgat-Cre *Ntn1* cKO mice displayed high extinction bursts (elevated pressing following reward omission) compared to controls that remained elevated on the second day of extinction training (*Figure 5E*). While these data likely reflect an altered motivational state with loss of Ntn1, it is also possible that the hyperactivity observed in Vgat-Cre *Ntn1* cKO mice contributes to the elevated lever press rates during FR5, PR, and extinction.

Analysis of sensory-motor gating in these mice revealed that Vgat-Cre mice injected with AAV1-FLEX-SaCas9-sg*Ntn1* had a significant reduction in the acoustic startle relative to control mice (*Figure 5F*) that was accompanied by a reduction in PPI (*Figure 5G*). Similar to mutagenesis of *Ntn1* in dopamine neurons, this manipulation in GABA neurons resulted in an increase in anxiety-like behavior as demonstrated by an increased time on edge; though we only observed a trend towards a reduction in time spent in the center of the open field arena (*Figure 5H*). The lack of observed significance in the time in center in the context of increased edge time may reflect the hyperactivity observed following *Ntn1* mutagenesis in VTA GABA neurons, consistent with this possibility, we did observe increased distance traveled during the open field test in these mice relative to controls (*Figure 5—figure supplement 1*).

### Loss of netrin-1 in dopamine neurons largely reverses the effects of *Ntn1* mutagenesis in GABA neurons

A loss of Ntn1 in VTA-dopamine neurons resulted in decreased excitatory synaptic input to those cells (theoretically reducing dopamine activity) (*Figure 6A*), and loss of Ntn1 in VTA-GABA neurons resulted in decreased excitatory tone onto GABA neurons, which would be predicted to increase dopamine activity through disinhibition (*Tan et al., 2012*; *Figure 6A*). Based on these observations, we asked whether a loss of *Ntn1* in both cell types would restore the balance of activity in the midbrain, or whether there is a hierarchical effect of *Ntn1* loss of function in GABA neurons. To address this, we

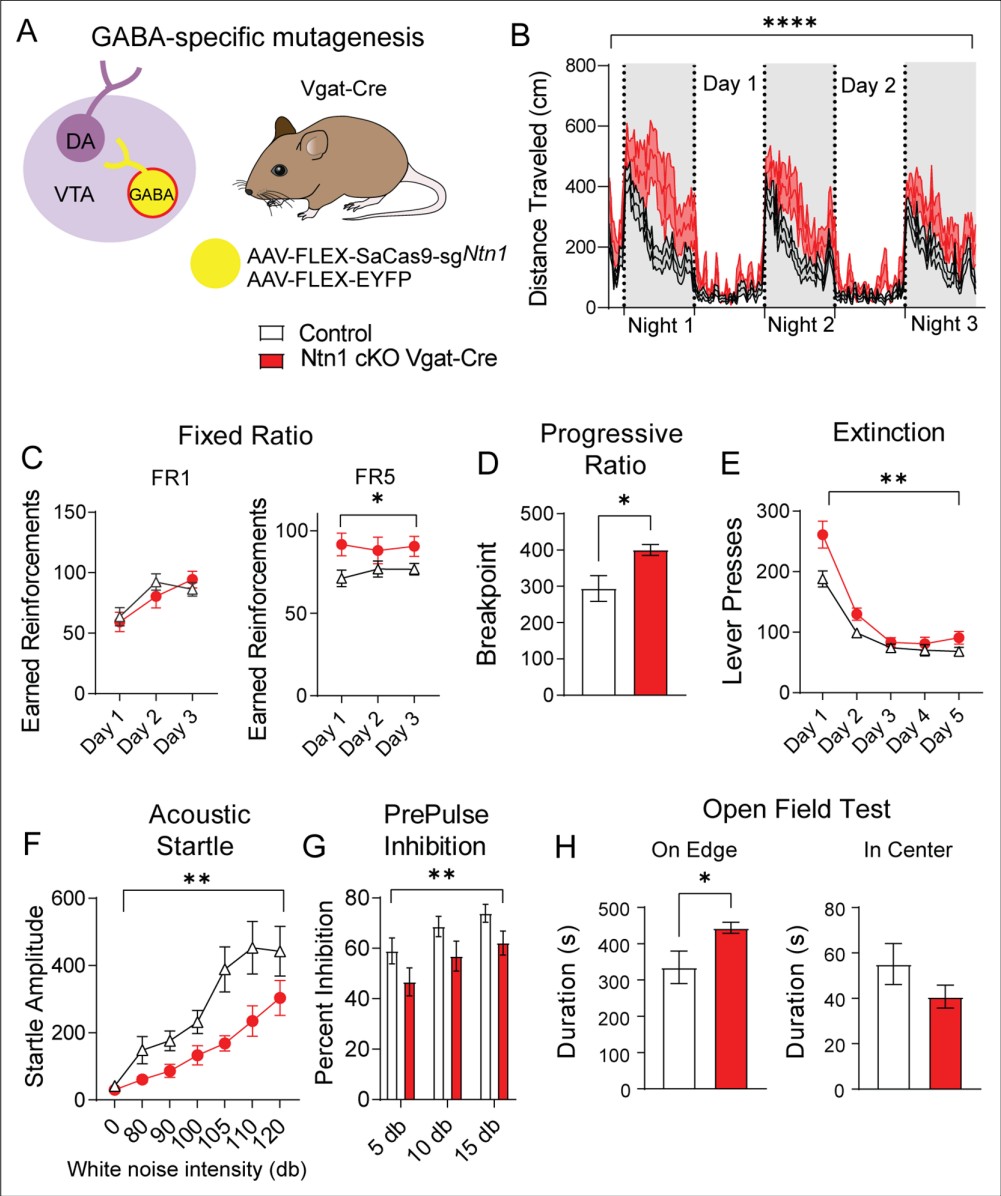

**Figure 5.** *Ntn1* cKO in GABA ventral tegmental area (VTA) neurons resulted in significant behavioral alterations. (**A**) Schematic summarizing cell type-specific knockout procedure. (**B**) Distance traveled in 15 min bins over the course of three nights and two days (n=26 controls, n=23 cKO, Two-way ANOVA Group $F_{(1, 11797)}$=527.4, ****$p<0.0001$, Time $F_{(250, 11797)}$=14.61 $p<0.0001$. Interaction $F_{(250, 11797)}$=1.342, $p=0.0003$). (**C**) Earned reinforcers during three days of FR1 or FR5 operant conditioning (n=18 control; n=15 cKO; FR1: Group $F_{(1, 31)}$=0.08647 $p=0.7707$, Time $F_{(2, 62)}$=30.46 $p<0.0001$, Interaction $F_{(2, 62)}$=3.186 $p=0.0482$; FR5: Group $F_{(1, 31)}$=4.261, *$p<0.05$, Time $F_{(1.992, 61.74)}$=0.3131 $p=0.7314$, Interaction $F_{(2, 62)}$=1.448 $p=0.2428$). (**D**) Breakpoint (maximum presses per reinforcer) on a progressive ratio task ($t=2.577$, df = 31, *$p<0.05$). (**E**) Lever presses per session during five days of extinction training (Group $F_{(1, 31)}$=10.23, **$p<0.01$, Time $F_{(1.491, 46.23)}$=83.84 $p<0.0001$, Interaction $F_{(4, 124)}$=3.546 $p=0.0089$). (**F**) Acoustic startle response to varying intensity white noise stimuli (Group $F_{(1, 31)}$=7.891, **$p<0.0085$, Intensity $F_{(1.790, 55.49)}$=24.94 $p<0.0001$, Interaction $F_{(6, 186)}$=2.186, $p=0.0462$). (**G**) Percent inhibition of startle response following pre-pulse at indicated intensities (Group $F_{(1, 93)}$=9.181, **$p<0.01$, Intensity $F_{(2, 93)}$=5.101 $p=0.0079$, Interaction $F_{(2, 93)}$=0.002227 $p=0.9978$). (**H**) Time on edge or in center of open field arena during a 10 min test session (edge: $t=2.248$, df = 31, *$p<0.05$, center $t=1.366$, df = 33, $p>0.05$).

The online version of this article includes the following source data and figure supplement(s) for figure 5:

**Source data 1.** Behavioral data for *Figure 5*.

*Figure 5 continued on next page*

*Figure 5 continued*

**Figure supplement 1.** Additional behavioral analysis of Ntn1 cKO Vgat-Cre mice.

**Figure supplement 1—source data 1.** Behavioral data for *Figure 5—figure supplement 1*.

crossed DAT-Cre with Vgat-Cre mice to develop a DAT-Cre::Vgat-Cre transgenic line, injected these mice with AAV1-FLEX-SaCas9-sg*Ntn1* or AAV1-FLEX-SaCas9-sg*Rosa26* (control) (*Figure 6B*), and assayed them using the previous behavioral battery.

Simultaneous *Ntn1* loss of function in VTA GABA and dopamine neurons largely reversed the hyperlocomotor phenotype (*Figure 6C*) observed with *Ntn1* mutagenesis in VTA GABA neurons alone, though a modest, increase in daytime locomotion remained (*Figure 6—figure supplement 1*). Similarly, loss of *Ntn1* in both VTA GABA and dopamine neurons resulted in operant responding during FR1 and FR5 that was similar to controls (*Figure 6D*) and pre-calorie restriction body weights did not differ between the groups. Motivation, as measured in the PR task, was elevated in the double transgenic Cre line following *Ntn1* mutagenesis (*Figure 6E*) and extinction was impaired (*Figure 6F*), though these phenotypes were less robust than those observed in the VTA GABA-only mice. Further analysis of extinction training days four and five revealed significant differences in both the number of lever presses and the rate of lever presses between groups (*Figure 6—figure supplement 1*). Finally, loss of Ntn1 in both cell types resulted in acoustic startle and PPI responses (*Figure 6G–H*), and open field activity (*Figure 6I*) that was similar to control mice.

## Discussion

Here, we show that Ntn1 is present in both dopamine and GABA-producing neurons of the adult VTA, and loss of Ntn1 function via genetic inactivation in either cell type results in a significant disruption of excitatory synaptic connectivity. The exact mechanisms by which Ntn1 regulates glutamatergic connectivity remain to be resolved. Likely mechanisms include Ntn1 regulation of the actin cytoskeleton and receptor transport vesicles through its activation of the cognate receptor DCC (*Yetnikoff et al., 2010*; *Rajasekharan and Kennedy, 2009*). The latter is consistent with our observed decrease in the amplitude of mEPSCs and reduced AMPA-evoked currents and with previous reports of Ntn1 regulating the delivery of GluA1-containing AMPA receptors to the postsynaptic density (*Jasmin et al., 2021*). Our finding that mEPSC frequency, but not paired-pulse ratio, was affected by Ntn1 loss further suggests netrin's role in modulating excitatory synaptic connectivity is likely confined to postsynaptic mechanisms.

Loss of Ntn1 in dopamine neurons had little effect on behavior; however, we did observe an increase in anxiety-like behavior as measured by the open field assay consistent with the proposed role of dopamine in the modulation of anxiety-related behavior (*Zarrindast and Khakpai, 2015*). The general lack of effect of reduced glutamatergic synaptic connectivity on appetitive behavior, locomotion, and sensory-motor gating is consistent with previous observations that reduced glutamatergic signaling in dopamine neurons largely does not affect these behaviors (*Zweifel et al., 2009*; *Hutchison et al., 2018*). However, this does not discount the importance of glutamatergic inputs to the VTA (see additional discussion below). In contrast, loss of Ntn1 in VTA GABA neurons had a significant effect on multiple behaviors including locomotion, motivation, and acoustic pre-pulse inhibition, all of which are consistent with a hyperdopaminergic phenotype and with previous reports that disrupting GABA neuron function in the VTA induces similar phenotypes (*Gore et al., 2017*; *Soden et al., 2020*).

Given the robust nature of the behavioral effects observed following *Ntn1* mutagenesis in VTA GABA neurons, we were initially surprised that simultaneous loss of Ntn1 in both GABA and dopamine neurons largely rescued the observed hyperdopaminergic phenotype. These results suggest that a balance of glutamatergic signaling in these two cell types is essential for the normal functioning of the mesolimbic dopamine system (*Figure 6A*). This finding is similar to what has been reported previously in the striatum.(*Beutler et al., 2011*) , demonstrated that loss of NMDA receptor signaling in dopamine D1 receptor-expressing neurons prevented the development of amphetamine sensitization; however, inactivation of NMDA receptors in D1R and D2R-expressing medium spiny neurons reversed this phenotype (*Beutler et al., 2011*). We have previously shown that blocking all synaptic transmission from GABA neurons in the VTA (*Gore et al., 2017*), or blocking selectively GABA release from VTA GABA neurons (*Hutchison et al., 2018*) results in hyperactivity and increased

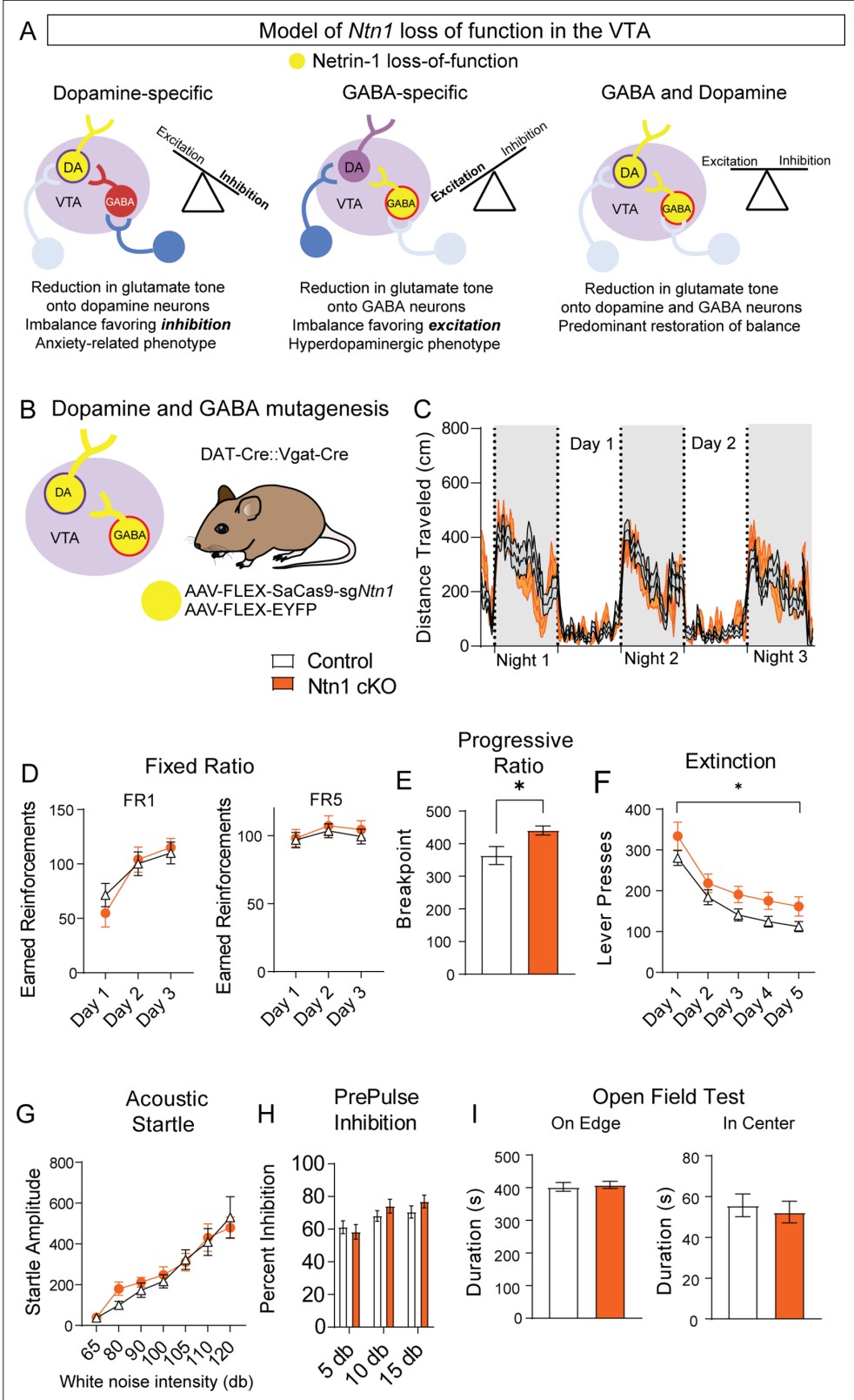

**Figure 6.** *Ntn1* cKO in DAT[IRES]::Vgat-Cre mice partially rescues behavioral phenotype. (**A**) Model of *Ntn1* loss of function in the ventral tegmental area (VTA) on excitatory and inhibitory balance. (**B**) Schematic of GABA and Dopamine Ntn1 cKO. (**C**) Distance traveled in 15 min bins over the course of three nights and two days (Two-Way ANOVA Group $F_{(1, 45)}=0.004273$, $p>0.05$, Time $F_{(17.16, 772.1)}=23.36$, $p<0.0001$, Interaction $F_{(247, 11115)}=1.492$,

*Figure 6 continued on next page*

*Figure 6 continued*

p<0.0001). (**D**) Earned reinforcers during three days of FR1 or FR5 operant conditioning (n=21 controls, n=20 Ntn1 cKO, FR1: Group $F_{(1, 40)}$=0.04247 p=0.8378, Time $F_{(2, 80)}$=25.70 p<0.0001, Interaction $F_{(2, 80)}$=1.402 p=0.2522; FR5: Group $F_{(1, 40)}$=0.2244 p=0.6383, Time $F_{(1.499, 59.95)}$=2.226 Pp0.1295, Interaction $F_{(2, 80)}$=0.1385 p=0.8708) (**E**) Breakpoint (maximum presses per reinforcer) on a progressive ratio task (t=2.502, df = 39, *p<0.05) (**F**) Lever presses per session during five days of extinction training (Group $F_{(1, 39)}$=6.990, *p=0.0117, Time $F_{(2.381, 92.87)}$=42.95 p<0.0001, Interaction $F_{(4, 156)}$=0.1470 p=0.9641). (**G**) Acoustic startle response to varying intensity white noise stimuli (Group $F_{(1, 40)}$=0.1207 p=0.7301, Intensity $F_{(1.775, 70.99)}$=36.77 p<0.0001, Interaction $F_{(6, 240)}$=0.6127 p=0.7201). (**G**) Percent inhibition of startle response following pre-pulse at indicated intensities (Group $F_{(1, 120)}$=0.9661 p=0.3276, Intensity $F_{(2, 120)}$=7.067 p=0.0013, Interaction $F_{(2, 120)}$=0.8861 p=0.4150). (**H**) Time on edge or in center of open field arena during a 10 min test session (edge: t=0.3584, df = 45 p>0,05, center: t=0.4233, df = 45, p>0.05).

The online version of this article includes the following source data and figure supplement(s) for figure 6:

**Source data 1.** Behavioral data for *Figure 6*.

**Figure supplement 1.** Average day and night locomotion in DAT-Cre::Vgat-Cre mice.

**Figure supplement 1—source data 1.** Behavioral data for *Figure 6—figure supplement 1*.

operant responding, though to a much greater degree than the effects observed here. We and others have also shown that loss of glutamate signaling in dopamine neurons has only a modest behavioral effect (*Zweifel et al., 2009*; *Hutchison et al., 2018*) so it is not surprising that reducing mEPSCs onto VTA dopamine neurons has little effect. The partial rescue of the GABA-Ntn1 knockout experiment suggests that reducing glutamatergic input onto dopamine neurons can abrogate effects associated with reduced inhibitory tone; thus, restoring the excitatory and inhibitory balance. Future research directly measuring the amount of inhibitory and excitatory current from the same cell (for example, by recording mIPSC at the reversal potential for EPSCs and mEPSCs at the reversal potential for mIPSCs) would be helpful in supporting this hypothesis, but was not performed in the current paper.

There are numerous disinhibitory projections to the VTA (*Soden et al., 2020*), and it is hypothesized that activation of GABAergic inputs onto VTA GABA neurons during behavior suppresses the activity of these neurons releasing the inhibitory brake onto dopamine neurons. Simultaneously, excitatory inputs onto VTA dopamine neurons, such as those from the pedunculopontine tegmental nucleus (*Soden et al., 2020*; *Lodge and Grace, 2006*), or dorsal raphe nucleus (*Qi et al., 2014*) would drive the glutamate-dependent burst activation of dopamine neurons (*Zweifel et al., 2009*; *Lodge and Grace, 2006*). Loss of excitatory drive onto dopamine neurons can likely be compensated for through the many disinhibitory circuits of the VTA. Indeed, inhibitory synaptic connectivity is not altered by the loss of *Ntn1*, and suppression of inhibition onto dopamine neurons is a potent means to drive the burst activation of these cells (*Paladini and Tepper, 1999*). The loss of glutamatergic inputs onto VTA GABA neurons is less likely to be bypassed at the circuit level, and we do not observe compensatory changes in excitatory or inhibitory synapses onto dopamine neurons following *Ntn1* mutagenesis in VTA GABA neurons or compensatory changes in inhibitory synaptic connectivity onto VTA GABA neurons. The effects of *Ntn1* loss of function in VTA GABA neurons can be abrogated if the excitatory drive onto dopamine neurons is also reduced by *Ntn1* mutagenesis. Excitatory inputs onto GABA neurons within the VTA proper, as well as the caudal tail of the VTA, or RMTg, arise from multiple locations including the lateral habenula and ventral pallidum (*Omelchenko et al., 2009*; *Brinschwitz et al., 2010*; *Tooley et al., 2018*). Activation of these circuits is aversive, likely through the activation of GABAergic inputs onto dopamine neurons (*Tooley et al., 2018*; *Wulff et al., 2019*; *Faget et al., 2018*; *Stamatakis and Stuber, 2012*). Thus, reduced glutamatergic inputs onto VTA GABA neurons would suppress the inhibition of dopamine neurons allowing excitatory drive onto these cells to go unchecked, resulting in the phenotypes observed here. When excitatory inputs onto dopamine neurons are also reduced the combined driving forces of disinhibition and excitation of the dopamine neurons are re-equilibrated, restoring balance to the system.

While our findings shed light on the role of Ntn1 in adult VTA neurons, the question remains as to which netrin-1 receptors may be involved. Indeed, though Dcc is considered to be its canonical receptor, Ntn1 is a known ligand for several additional receptors, including DSCAM, Neogenin, and Unc5 homologs A-D (*Rajasekharan and Kennedy, 2009*; *Lai Wing Sun et al., 2011*). Previous work has identified the presence of both Dcc and Unc5c receptors in the adult VTA (often in the same

cells) (*Manitt et al., 2010*). It is also interesting to note that during the development of the spinal cord, Ntn1 expression in the floor plate attracts commissural axons to the midline, but following the arrival of these axons at the floor plate, Unc5 expression increases to suppress the attractive actions of DCC signaling (*Lai Wing Sun et al., 2011*). Whether a similar relationship exists for the formation of nascent synapses and the maintenance of excitatory synapses occurs in the VTA will be important to resolve. Of further note, in addition to the role of Ntn1/Dcc/Unc5 signaling in the regulation of commissural axons crossing the midline, Slit/Robo signaling repels axons away from the floor plate (*Kidd et al., 1998*; *Kidd et al., 1999*) setting up a push-pull relationship between these pathways. We previously demonstrated that Robo2 maintains inhibitory synaptic connectivity in the adult VTA (*Gore et al., 2017*), suggesting the existence of another 'push/pull' relationship between these two pathways in which netrin/Dcc/Unc5 regulates excitation and Slit/Robo signaling regulates inhibition.

We find that *Ntn1* is expressed in both Vgat- and Th-expressing neurons which is consistent with single nuclear RNA sequencing data recently obtained in rats from *Phillips et al., 2022*, though the levels of nuclear *Ntn1* mRNA levels in this study were relatively low. This may reflect differences between rat and mouse, or the increased sensitivity of in situ hybridization in detecting cytosolic mRNA compared to nuclear mRNA levels observed with snRNA seq. Interestingly, (*Phillips et al., 2022*) also observed the expression of *Ntn1* in glial cells. Thus, the identity and potential role of the roughly 10% of *Ntn1* expressing neurons that do not co-label with markers of dopamine or GABA neurons in the VTA, potentially glutamatergic neurons or glial cells, warrants further investigation.

In our proposed model, reduced excitatory input onto GABA neurons results in a reduced inhibition of dopamine neurons causing the observed phenotypes, consistent with previous observations that reducing VTA GABA neuron function causes hyperactivity (*Gore et al., 2017*; *Soden et al., 2020*). However, it is also likely that GABA neurons projecting outside the VTA (for example, to the NAc *Brown et al., 2012*) are also affected and could contribute to the observed behavioral phenotypes.

Mutations in *NTN1* (netrin-1) and *DCC* in humans have been associated with several dopamine-associated psychiatric conditions, including neurodevelopmental disorders such as schizophrenia (*Wang et al., 2018*; *Tang et al., 2019*; *Grant et al., 2012*) and major depressive disorder (*Tang et al., 2019*; *Zeng et al., 2017*; *Vosberg et al., 2020*; *Torres-Berrío et al., 2020*), as well as multiple neurodegenerative disorders (*Lesnick et al., 2007*; *Lesnick et al., 2008*; *Lin et al., 2009*). Our findings that *Ntn1* plays a key role in maintaining excitatory connectivity in the adult midbrain and controlling the inhibitory/excitatory balance in this region highlights the importance of understanding these critical developmental signaling pathways in the adult nervous system that are likely important for therapeutic considerations in targeting these pathways.

## Methods

### Mice

All procedures were approved and conducted in accordance with the guidelines of the University of Washington's Institutional Animal Care and Use Committee (protocol number: 4249–01). Mice were housed on a 12:12 light:dark cycle with *ad libitium* access to food and water, except when undergoing food restriction for operant behavioral conditioning. Approximately equal numbers of male and female mice were used for each experiment. Post-hoc analysis to test for sex-specific differences was performed for each experiment and no differences were observed so mice from each sex were pooled within experimental and control groups. Mice were group housed (2–5 mice per cage, separated based on sex at the time of weaning). Mice injected with CRISPR/YFP were allowed 4–5 weeks of recovery after surgery to allow for viral expression, mutagenesis, and protein turnover before any testing. Dat-Cre ($Slc6a3^{Cre/+}$) mice and Vgat-Cre ($Slc32a1^{Cre/+}$) mice were obtained from Jackson Laboratories (Strain # 006660 and 028862).

### Viruses

All adeno-associated viruses (AAV) for CRISPR/SaCas9 mutagenesis were produced in-house, as previously described (*Hunker et al., 2020*). CRISPR viruses employed for this research: AAV1-FLEX-SaCas9-U6-sgNtn1 (Addgene: #159907) and AAV1-FLEX-SaCas9-U6-sgRosa26 (Addgene: #159914) are available through Addgene or upon request to the corresponding author.

## Surgeries

All mice used were 8–10 weeks of age at the time of surgery. Mice were induced using isoflurane at 5.0% and held at 2% throughout the procedure. Mice were stereotaxically injected bilaterally into the VTA using the following coordinates in mm, relative to bregma: A/P: –3.25; M/L±0.5; D/V: (–4.9) – (–4.4), total volume 0.5 µL into each side. A/P coordinates were adjusted for Bregma/Lambda distances using a correction factor of 4.2 mm.

## In situ hybridization

Male and female mice (n=2 each sex, 8–12 weeks old) were used to verify mRNA expression in the VTA using RNAscope. Brains were flash-frozen in 2-methylbutane and representative coronal sections that spanned the VTA were sliced at 20 µm and slide mounted for hybridization. Sections were prepared for hybridization per the manufacturer's (Advanced Cell Diagnostics, Inc) instructions using probes for *Th* (Mm-*Th*), *Ntn1* (Mm-*Ntn1*-C2), and *Slc32a1* (Vgat; Mm-*Slc32a1*-C3). Slides were coverslipped with Fluoromount with DAPI (Southern Biotech) and imaged using a confocal fluorescent microscope (the University of Washington Keck Center Leica SP8X confocal) and Keyence Fluorescence Microscope (Keyence). Quantification of co-labeled cells was performed using CellProfiler, with thresholding and cell identification/overlap for each channel verified for each image manually prior to quantification.

## Immunohistochemistry

Mice were anesthetized with pentobarbital and transcardially perfused with PBS followed by 4% PFA. Brains were post-fixed for 24 hr in PFA at 4°C, followed by 48 hr in 30% sucrose. The VTA was coronally sectioned at 30 µm. Sections were kept in PBS with 0.3% Sodium Azide. Free-floating sections were treated with 0.3% TBS-Triton-X 100 3x10 min, blocked in 3% Normal Donkey Serum for 1 hr and treated overnight in primary antibody. Following 1–3 hr in secondary antibody (JacksonImmuno), sections were slide mounted and coverslipped with Fluoromount with DAPI. Images were collected on a Keyence Fluorescence Microscope (Keyence). For CRISPR validation, male and female DAT-Cre mice (8–12 weeks old) received AAV1-FLEX-SaCas9-U6-sgNtn1/AAV1-FLEX-YFP (Ntn1-cKO) or AAV1-FLEX-SaCas9-U6-sgROSA26/AAV1-FLEX-YFP (controls) injections as described above, and quantification of co-labeled cells for immunohistochemistry were performed using ImageJ 1.53 Cell Counter/Multi-point tool. Total Th-positive cells were recorded for all images and averaged across all slices for each mouse to give a total number of Th-positive cells. Primary antibodies used: mouse anti-TH (1:1500, Millipore), chicken anti-Netrin-1 (1:1000, Abcam), and rabbit anti-HA (1:1500, Sigma).

## Slice electrophysiology

Mice injected with CRISPR/YFP were allowed 4–5 weeks of recovery after surgery to allow for viral expression, mutagenesis, and protein turnover. All solutions were continuously bubbled with $O_2$/$CO_2$. Horizontal (200 µm) brain slices were prepared from 12 to 20 week-old mice in a slush NMDG cutting solution (*Lin et al., 2009*) (in mM: 92 NMDG, 2.5 KCl, 1.25 $NaH_2PO_4$, 30 $NaHCO_3$, 20 HEPES, 25 glucose, 2 thiourea, 5 Na-ascorbate, 3 Na-pyruvate, 0.5 $CaCl_2$, 10 $MgSO_4$, pH 7.3–7.4). Slices recovered for ~12 min in the same solution warmed in a 32 °C water bath, then transferred to room temperature HEPES-aCSF solution (in mM: 92 NaCl, 2.5 KCl, 1.25 $NaH_2PO_4$, 30 $NaHCO_3$, 20 HEPES, 25 glucose, 2 thiouria, 5 Na-ascorbate, 3 Na-pyruvate, 2 $CaCl_2$, 2 $MgSO_4$). Slices recovered for an additional 30–60 min in HEPES solution at room temp. Whole-cell patch-clamp recordings were made using an Axopatch 700B amplifier (Molecular Devices) using 3–5 MΩ electrodes. Recordings were made in aCSF (in mM: 126 NaCl, 2.5 KCl, 1.2 $NaH_2PO_4$, 1.2 $MgCl_2$, 11 D-glucose, 18 $NaHCO_3$, 2.4 $CaCl_2$) at 32 °C continually perfused over slices at a rate of ~1 ml/min. VTA dopamine and non-dopamine neurons were identified by fluorescence.

### mE/IPSC

For miniature excitatory postsynaptic currents (mEPSCs), the internal solution contained: 130 mM K-gluconate, 10 mM HEPES, 5 mM NaCl, 1 mM EGTA, 5 mM Mg-ATP, 0.5 mM Na-GTP. Picrotoxin (200 µM) was added to ACSF to block $GABA_A$ receptor-mediated events. For miniature inhibitory postsynaptic currents (mIPSCs), the internal solution contained: 135 mM KCl, 12 mM NaCl, 0.05 mM EGTA, 100 mM HEPES, 0.2 mM Mg-ATP, 0.02, and Na-GTP mM. To block glutamatergic events, 2 mM kynurenic acid was bath applied in the ACSF. All mIPSCs and mEPSCs cells were recorded in the

presence of 1 mM tetrodotoxin (TTX) to block action potentials. Cells were held at −60 mV for a minimum of 5 minprior to data acquisition. Data were analyzed using Clampfit 10.3 (pCLAMP 11 Software Suite, Molecular Instruments).

## Paired pulse ratio

For PPR, the internal solution contained: 130 mM K-gluconate, 10 mM HEPES, 5 mM NaCl, 1 mM EGTA, 5 mM Mg-ATP, 0.5 mM Na-GTP. Picrotoxin (200 µM) was added to ACSF to block GABA$_A$ receptor-mediated events. Electrical stimulation was delivered using a concentric bipolar electrode placed rostral to the VTA. Data were analyzed using Clampfit 10.3 (pCLAMP 11 Software Suite, Molecular Instruments).

## Bath AMPA and NMDA application

For AMPA currents, neurons were held at −60 mV and 50 µM cyclothiazide was perfused onto the slice for 30 s, followed by 1 µM AMPA (with cyclothiazide) for 30 s. For NMDA currents, neurons were held at + 40 mV and 50 µM NMDA was perfused onto the slice for 30 s. Picrotoxin (100 µM) and tetrodotoxin (500 nM) were included in the bath.

## **Behavior**

### Locomotor activity

Four weeks after surgery, baseline locomotion was measured using locomotion chambers (Columbus instruments) that use infrared beam breaks to calculate ambulatory activity. Mice were singly housed in Allentown cages with reduced corncob bedding and provided with ad libitum access to food and water. Locomotion was monitored continuously for three nights two days.

### Open field testing

Mice were placed in a large circular arena (120 cm diameter) and activity was recorded for a period of 10 min using Ethovision software. Zones signifying arena edge and center were generated for each video using Ethovision and kept consistent in size and placement across all trials. Time in center, time on edge, and total distance were calculated. Thigmotaxis, the tendency of mice to remain in close proximity to the walls of an enclosure, has been validated as a measure of anxiogenic behavior (*Simon et al., 1994*; *Seibenhener and Wooten, 2015*), with increased time on edge signifying increased anxiety-like behavior.

### Operant conditioning

Prior to testing, all mice were food restricted to 85% body weight and maintained at this level throughout operant conditioning. Mouse weights are included in *Figure 4—figure supplement 1*, *Figure 5—figure supplement 1*, *Figure 6—figure supplement 1*. Mice were tested on an operant conditioning paradigm in Med Associates boxes in the following order: FR1, FR5, Progressive Ratio, Reinstatement, and Extinction. Each fixed ratio 1 (FR1) session lasted for 60 min. Levers were extended and remained extended until a lever press. Upon a lever press, levers were retracted and a sucrose pellet was immediately delivered into the food hopper. The levers did not extend again until the mouse made a head entry into the food hopper to retrieve the pellet. Reinforced FR1 sessions lasted for three days, followed by three days of FR5 (five lever presses required to obtain sucrose pellet), and a single day of a progressive ratio where the number of lever presses necessary for sucrose pellet delivery increases non-arithmetically (i.e. 1, 2, 4, 6, 9, 13…) over the course of the session. The progressive ratio session ended after three consecutive min of no lever presses or after 3 hr. After progressive ratio, mice again underwent FR1 reinforced training, followed by extinction for 60 min each session for five days. Here, levers extend and retract similarly to the FR1 reinforced paradigm, yet a sucrose pellet reward is omitted.

### Acoustic startle and prepulse inhibition

Acoustic startle responses were measured using acoustic startle chambers (San Diego Instruments). Prior to testing mice received a 10 min habituation period. Background noise was maintained at 65 dB throughout testing. After habituation, mice were presented with 5, 40 ms duration 120 dB,

pulse-alone trials to obtain baseline startle responses, followed by 50 trials of either a startle pulse-alone, 1 of 3 prepulse trials, or a null trial, in which no acoustic stimulus is presented. Startle trials consisted of a 40 ms, 120 dB pulse of white noise. The three prepulse trials consisted of a 20 ms prepulse of 70-, 75-, or 80 dB intensity (5, 10, and 15 dB above background) that preceded 120 dB startle pulse by 100 ms. Peak amplitude of the startle response (65 ms after pulse onset) was used as the measure of startle response magnitude.

## Sex differences

No sex differences were observed in any of the behavioral or electrophysiological results.

## Statistics

Data were analyzed for statistical significance using GraphPad Prism. All statistical tests were two-sided and corrected for multiple comparisons where appropriate. All experiments were repeated at least twice using a double-blind design.

## Materials availability

All newly created reagents are freely available upon request to the corresponding author.

## Acknowledgements

We would like to thank the staff of the University of Washington's Comparative Medicine Animal Facilities, the University of Washington's Keck Imaging Center, and the administrative staff of the Molecular and Cellular Biology Graduate Program.

This study was supported by grants from the National Institutes of Health T32GM007270 (MC), 1F31MH126489-01A1 (MC), T32DA727825 (B.J.), K99DA054265 (B.J.), R01MH104450 (LSZ), and. R01DA044315 (LSZ). B.J., Ph.D., holds a Postdoctoral Enrichment Program Award from the Burroughs Wellcome Fund. We would also like to acknowledge support from the University Of Washington Center Of Excellence in Opioid Addiction Research/ Molecular Genetics Resource Core (P30DA048736). The authors declare no conflicting interests.

## Additional information

### Funding

| Funder | Grant reference number | Author |
| --- | --- | --- |
| National Institutes of Health | R01DA044315 | Larry S Zweifel |
| National Institutes of Health | R01MH104450 | Larry S Zweifel |
| National Institutes of Health | 1F31MH126489 | Marcella M Cline |
| National Institutes of Health | K99DA054265 | Barbara Juarez |

The funders had no role in study design, data collection and interpretation, or the decision to submit the work for publication.

### Author contributions

Marcella M Cline, Conceptualization, Data curation, Formal analysis, Funding acquisition, Validation, Investigation, Methodology, Writing – original draft, Writing – review and editing; Barbara Juarez, Investigation, Methodology; Avery Hunker, Formal analysis, Methodology; Ernesto G Regiarto, Bryan Hariadi, Investigation; Marta E Soden, Data curation, Formal analysis, Methodology; Larry S Zweifel, Conceptualization, Resources, Data curation, Formal analysis, Supervision, Funding acquisition, Validation, Investigation, Methodology, Writing – original draft, Project administration, Writing – review and editing

## Author ORCIDs
Marcella M Cline  http://orcid.org/0000-0002-9590-3725
Larry S Zweifel  http://orcid.org/0000-0003-3465-5331

## Ethics
All procedures were approved and conduced in accordance with the guidelines of the University of Washington's Institutional Animal Care and Use Committee, protocol number 4249-01.

## Decision letter and Author response
Decision letter https://doi.org/10.7554/eLife.83760.sa1
Author response https://doi.org/10.7554/eLife.83760.sa2

---

# Additional files

## Supplementary files
• MDAR checklist

## Data availability
All data generated or analysed during this study are included in the manuscript and supporting file.

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
