## [Editor Report]

This manuscript reports an important, previously unappreciated, non-developmental role for the guidance cue netrin-1 in midbrain physiology and related behavior in adult animals. Using multiple experimental tools in adult mice, the study convincingly shows that netrin-1 within midbrain dopamine and GABA neurons is necessary to maintain dopamine excitatory tone and plays a role in motivated and anxiety-like behavior. This paper will be of interest to neuroscientists studying dopamine function and/or motivated behavior and those interested in ways that neurodevelopmental genes can continue to play a role in neuronal function and behavior into adulthood.

---

## [Decision Letter]

**Decision letter after peer review:**

[Editors’ note: the authors submitted for reconsideration following the decision after peer review. What follows is the decision letter after the first round of review.]

Thank you for submitting the paper "Netrin-1 regulates the balance of glutamatergic connectivity in the adult ventral tegmental area" for consideration by *eLife*. Your article has been reviewed by 3 peer reviewers, and the evaluation has been overseen by a Reviewing Editor and a Senior Editor. The reviewers opted to remain anonymous.

Comments to the Authors:

We are sorry to say that, after consultation with the reviewers, we have decided that this work will not be considered further for publication by *eLife*.

We agreed that the work addresses an interesting question and that the manipulation to knockdown netrin-1 in adult VTA was elegant. However, we noted several concerns. These include unaddressed issues related to the precise role of netrin function in DA neurons leading to challenges in interpreting the behavioral results, the lack of mechanistic insight into the observed effects of glutamate, and contextualization in prior literature examining netrin's effects in the VTA. Addressing the concerns is likely to be well beyond the scope of an *eLife* revision (which under normal circumstances is limited to 2 months).

*Reviewer #1 (Recommendations for the authors):*

The study seeks to investigate whether the guidance cue netrin-1 in the VTA is involved in the regulation of glutamatergic synaptic connectivity of dopamine and GABA neurons and influences behaviors known to be mediated by mesolimbic dopamine systems.

The study involves multiple approaches: viral mediated inactivation of NTn1 in VTA GABA or dopamine neurons, using Cre inducible CRISPR/Cas9; in situ hybridization to confirm colocalization of Ntn1 and tyrosine hydroxylase or Ntn1 and SlC31a1; electrophysiological recordings to evaluate changes in neural excitability of dopamine or GABA neurons, a battery of behavioral tests to evaluate whether loss of Ntn1 in either cell type alters sensorimotor gating function and motivated and anxiety-like behaviors.

Major strengths of the paper include (i) the novelty of the question addressed and the multidisciplinary approach, (ii) most of the experiments address functional questions, (iii) the careful characterization of functional and behavioral phenotypes for the majority of the experiments, (iv) the generation of a mechanistic model potentially explaining how netrin-1 in the adult VTA, by regulating the excitatory tone of dopamine and GABA neurons, regulates overall dopamine excitability and behavior.

Weaknesses of the study are mainly related to the interpretation of their results and the conclusions made: (i) Regarding the results from the in situ hybridization experiment, the authors do not compare their findings with results from a recent single-cell RNA seq study (PMID: 35385745). There is a discrepancy between the findings, particularly in the proportion of GABA neurons expressing NTn1 (as well as in the expression of NTn1 in other cell types, such as astrocytes). The authors do not mention this issue in their results or discussion. (ii) The authors conclude that their findings demonstrate that netrin-1 regulates the balance of glutamatergic connectivity in the VTA (title, last paragraph introduction, results, and conclusion). Indeed they find that downregulation of Ntn1 in VTA dopamine or GABA cells reduces the frequency and amplitude of their miniature excitatory postsynaptic currents (mEPSCs), without altering miniature inhibitory postsynaptic currents (mIPSCs) or leading to statistically reliable changes in frequency or amplitude in other (non-specified) cell types. They also show that reduced mEPSCs is not mediated by changes in presynaptic release, as revealed by paired-pulse ratio measures. However, these findings are not directly linking electrophysiological changes to alterations in glutamatergic synaptic connectivity (iii) Whether NTn1 deletion leads to dopamine or GABA neuronal loss remains unknown. This is important considering previous studies linking (or not) changes in the netrin-1 system in VTA neurons and cell loss. (iv) The study shows that downregulation of Ntn1 in dopamine neurons has no significant effect on reward but influences anxiety-like behaviors. In contrast, downregulation of Ntn1 in GABA neurons produces changes in most of the behaviors tested. It is not clear, however, whether increased locomotor activity in mice with Ntn1 deletion in GABA neurons could influence changes in lever pressing in the operant behaviors. (v) Regarding the changes in extinction training in the mice with reduced Ntn1 in GABA neurons, it seems that they are lever pressing at a higher rate from the beginning but they extinguish their behavior at a similar rate (similar for the animals with reduced Ntn1 in GABA and in dopamine neurons). (vi) the model and hypothesis put forward and tested in the experiments shown in Figure 6 are very interesting. However, the results obtained do not justify the conclusion that Loss of Ntn1 function in both cell types simultaneously largely rescues the consequences induced by GABA- only Ntn1 deletion.

To be able to link the electrophysiological changes to glutamatergic synaptic connectivity, other experiments are required, including assessing structural changes (e.g. PMID: 24174661) in dopamine and GABA neurons as well as the proportion of AMPA/NMDA receptors and AMPA/NMDA ratios. In this regard, there is a previous study relating the netrin-1 guidance cue system in adult VTA synaptic plasticity (PMID: 20345916).

Whether NTn1 deletion leads to dopamine or GABA neuronal loss could be addressed using stereology.

To be able to conclude that the loss of Ntn1 function in both cell types simultaneously largely rescues the consequences induced by GABA- only Ntn1 deletion, the electrophysiological properties of dopamine (and also GABA) neurons could be assessed. This experiment will also test more directly the model the authors are proposing.

In the discussion, evidence showing the role of glutamatergic inputs of VTA dopamine neurons on behavior needs to be revised more carefully (e.g. PMID 25388237, PMID 26631475, and PMID 30699344).

*Reviewer #2 (Recommendations for the authors):*

This manuscript by Cline et al. sought to define the role of the axonal guidance cue netrin-1 in synaptic signaling in the ventral tegmental area (VTA). The authors used CRISPR-Cas9 mutagenesis to reduce netrin-1 expression/function in the two predominant neuronal types in the VTA, dopaminergic and GABAergic neurons. This work builds on previous work from this group showing a role for the axon guidance receptor ROBO2 in inhibitory connectivity in adult VTA. Netrin-1 is examined here because of its persistent expression in the VTA into adulthood, despite its established role as a developmental protein. A strong combination of techniques is used, including selective knockdown of the netrin-1 gene in multiple neuron types in the VTA, patch clamp electrophysiology, and several behavioral assays. The results clearly indicate that knockdown of netrin-1 specifically in either GABA or dopamine neurons reduces miniature glutamatergic synaptic currents specifically in that cell type. Interestingly, inhibitory input was not significantly affected. This is consistent with previous reports of effects on excitatory synapses in the hippocampus. Robust behavioral consequences were only reported in the GABA neuron knockdown and were not evident when netrin-1 was knocked down in both cell types. The authors conclude that netrin-1 is important for maintaining the balance of excitatory input onto the two main neuronal subtypes in the VTA. This conclusion is largely supported by the results from the experiments, which were performed rigorously. The manuscript itself was easy to follow and well written, save for some minor omissions.

Operant responding for food was used as one of the dependent measures in Figures 4 and 5, with differences observed in the GABA neuron knockdown, however, one omission of the study was that body weights of the mice before and especially after treatment were not reported. It may be that viral knockdown of netrin-1 in one cell type has effects for instance on satiety, producing effects on behavior along with energy balance. The addition of body weight data would help round out the data set and, if different, might help with interpretation. Details were also not provided for the food restriction procedure that was used during operant conditioning for food pellet responding, which could have further interacted with the netrin-1 manipulation in the mouse lines to affect physiology, behavior, or both.

As discussed above, most of this study is strong, with interpretations supported by largely convincing results. However, the manuscript could be improved with additions and clarifications.

While both cell types showed increased mEPSC frequency and amplitude after netrin-1 knockdown, only the GABA neuron knockdowns showed robust behavioral effects, including increased locomotion (day and night), and increased operant responding for food, and decreased acoustic startle and pre-pulse inhibition. As some of the behavioral tasks involve responding for food pellets, interpretation of the results would benefit from reporting the weights of the mice before and after treatment. Additionally, details about the food restriction procedure that was used for operant conditioning should be provided. Several reports have identified the effects of the feeding state on dopamine neuron excitability and synaptic input to the area. Was the food restriction controlled to a percent loss of body weight, applied acutely or chronically, done throughout the operant study or just during FR1, etc.? Did the mice always eat the pellets when performing the operant task? It's interesting that responding in all of the dopamine mice stayed at the same values when they switched from FR1 to FR5, but in the GABA mice, the control animals fell while the netrin-1 knockdowns stayed the same. Full disclosure of these details would help the reader interpret small observations in the data and later assist in reproducing the results, should they wish to do so.

The evidence for the anxiety behavioral phenotype in the dopamine neuron-specific netrin-1 knockdown is pretty thin, as a time-in-center measure in an open field could be affected by other factors, and other supporting data for instance from elevated mazes were not used. Admittedly, anxiety can be difficult to show in mice. However, Reference 20 which was used to support the "proposed role of dopamine in the modulation of anxiety-related behavior" was a dead-end non-citation. If the authors wish to keep this part of their interpretation they should bolster the explanation of the link between dopamine and anxiety with further evidence (experimental or literature). Also, details about the meaning of "time in center" and "time on edge" should be provided in the Methods.

The extinction result in the GABA mice is presented as "a significant delay in the rate of extinction following reinstatement of FR1." This is hard to interpret because responding immediately before extinction is not given and the data are presented only as raw numbers instead of percent of baseline. If the netrin-1 knockdowns were responding more at FR1 when they were returned to that condition, the shape of the curve would actually indicate no difference in extinction. The same could be true in Figure 6 with the double cell knockdowns. Depending on what the data look like, these graphs may be more accurately presented as normalized numbers. This is a minor issue in the scheme of this paper but it should nonetheless be clarified.

The schematics in Figure 6A show that the effects on the GABA neurons proceed through the dopamine neurons. While this is entirely plausible, the authors never actually show this experimentally, for instance by locally manipulating GABA input in virus-injected mice. It may be at least as likely that the important interactions occur in the nucleus accumbens, or some other area to which both cell types project. As a minimum, the authors should point out this caveat in the Discussion, or point out any other possibility that could also explain the somewhat surprising data in Figure 6.

There's some confusion about the Ntn1 co-localization in Figure 1. The language in the caption and Figure 1 itself seem clear. However, the text on page 3 seems to contradict the language in the figure caption. The way it is worded, instead of 64, 30, and 6% shouldn't these numbers be 72, 18, and 10%? Please clarify with the correct information, or point out where the confusion lies.

In the abstract, "simultaneously" is placed confusingly in the sentence about rescuing the GABA phenotype. This would be more clear if the sentence started "Simultaneous loss… in both cell types."

Figure 1D has two (identical) scale bars.

*Reviewer #3 (Recommendations for the authors):*

In this study, they used genetic strategies to decrease Netrin-1 expression in either dopamine or GABA neurons of the VTA. Reduction of Netrin-1 expression in VTA dopamine neurons decreased excitatory, but not inhibitory postsynaptic currents onto TH^+^ or GABA VTA neurons. While the loss of netrin-1 in dopamine neurons did not significantly influence behaviour, loss of netrin-1 in GABAergic neurons increased locomotor activity, effort for rewards, extinction delay, and decreased prepulse inhibition. Finally, effects on locomotor activity, reward seeking, and prepulse inhibition observed in GABA-netrin mice were not present when there was the loss of netrin-1 in both dopamine and GABA neurons, suggesting that loss of netrin-1 in both could restore the behavioural effects of loss of netrin-1 in GABA neurons.

Major strengths:

This is a nicely written, clearly illustrated study describing the loss of netrin-1 in VTA dopamine neurons and GABA neurons. The authors use an elegant genetic methodology to knock down netrin-1 in select populations of neurons within the VTA. They use a battery of behavioural assays to examine the effects of this knockdown.

Major weaknesses:

While this study provides a potential physiological mechanism underlying the changed behavioural effects, it doesn't connect these changes to the behaviour or identify the mechanism by which adult netrin-1 influences these changes in synaptic transmission. While netrin-1 is involved in synapse formation during development, it is not clear how netrin-1 is influencing synapses in adulthood. For example, is it necessary for the stabilization of synapses?

Author suggestions:

The authors propose that loss of netrin-1 in dopamine neurons leads to enhanced excitation and decreased inhibition, whereas loss in GABA neurons would lead to enhanced inhibition and decreased excitation, and the E:I ratio would be balanced when netrin is lost from both cell population. However, they do not test this assertion by measuring excitatory:inhibitory ratio in each model. This would support their hypotheses in figure 6.

Some of the discussion focuses on the role of netrin-1 in development. However, the manipulations done in this paper were to remove netrin-1 in adulthood after axon migration and synapse formation occur. They do not discuss what the 'adult function' of netrin-1 is. While it seems to play a role in excitatory synaptic transmission, given its effects on mEPSCs, the authors did not provide sufficient information to conclude if this was a reduction in the number of synapses, a silencing of synapses, or a decrease in release probability with compensatory postsynaptic changes. Experiments addressing how adult netrin-1 signaling in the VTA specifically influences synaptic transmission onto dopamine or GABA neurons of the VTA may highlight how netrin-1 might be contributing to associated behavioral changes.

[Editors’ note: further revisions were suggested prior to acceptance, as described below.]

Thank you for resubmitting your work entitled "Netrin-1 regulates the balance of glutamatergic connectivity in the adult ventral tegmental area" for further consideration by *eLife*. Your revised article has been evaluated by Kate Wassum (Senior Editor) and a Reviewing Editor.

The manuscript has been improved but there are some remaining issues that need to be addressed, as outlined below:

Essential revisions:

We have a few remaining concerns that can be addressed with an additional analysis of existing data (see R3 point #1) and edits to the text to add clarity, change language, temper conclusions, discuss alternatives, etc.

Please provide a point x point response to each reviewer point along with your revision.

*Reviewer #1 (Recommendations for the authors):*

This resubmitted manuscript is substantially improved from the previous version. Additions and clarification to methods and interpretations have largely addressed my previous concerns, which were minor. Additional discussion has bolstered the notion that netrin-1 is impotant for maintaining the balance of excitatory and inhibitory inputs to dopamine neurons. The addition of maze data would have bolstered conclusions about the anxiety phenotype, but "time on edge" and locomotor data was added and this was always a minor point that did not detract substantially from the rest of the manuscript.

One item does need to be clarified. The experiment represented in Figure 3 Supplement 3 was performed to address a concern of Reviewer 1, however the text describing this is very confusing. The Results describe these data as evoked EPSCs following bath perfusion of AMPA or NMDA. The data themselves seem to be showing changes in holding current in a voltage clamp experiment, although the axis labels and the lack of sample traces leave this in doubt. If it is holding current, more details should to be provided in the figure (including better axis labels and a description of how long the drug went on and the holding voltage, both of which are currently only in the Methods). If instead it is evoked EPSCs in the presence of bath perfused agonists, then other details also need to be provided to make this result make sense (such as sample traces and an explanation of what they were after).

The addition of body weight data and food restriction information is appreciated. However in the Response to Reviewers the level of restriction was listed as 85% of initial body weight, whereas in the manuscript this is given as 80%. Please make sure the correct number appears in the manuscript.

*Reviewer #2 (Recommendations for the authors):*

The revised version improved the interpretation of the results, and I am pleased that the authors replied to most of my comments on the previous version. They added needed citations, clarification on their manipulation, and additional experiments to understand the changes in synaptic connectivity better and clarify the electrophysiological changes that occur with Ntn1 mutagenesis, as well as their proposed model.

The following issues remain unresolved:

– Since the study does not assess alterations in neuronal structure and connectivity, we suggest the word "connectivity" to be dropped or modified.

– It appears that stereology was not performed to calculate the number of neurons expressing netrin-1 in TH^+^ or GABA+ cells in the mice with conditional netrin-1 KO. The statement "Total Th-positive cells were recorded for all images and averaged across all slices for each mouse to give a total number of Th-positive cells" needs to be revised, because total number of cells can only be assessed using stereological analysis. It remains unknown whether NTn1 deletion leads to dopamine or GABA neuronal loss. This is an important issue that needs to be acknowledged in the manuscript. Please see

https://www.embopress.org/doi/full/10.15252/embj.2020105537

– Regarding the increase in locomotor activity observed after the downregulation of Ntn1 in GABA neurons, the authors argue that the level of responding in the FR1 task is not altered in GABA-Ntn1 mutant mice, suggesting that elevated responding in the FR5 and PR assays is not likely a reflection of hyperactivity. Yet, increasing the response rate in the FR1 protocol is no possible. Once the animals respond to the reward in the FR1, the lever retracts, and no more lever-pressing is possible until a reward is collected. The statement added by the authors "however, we did not observe differences in operant responding during FR1 training or reinstatement indicating that the observed effects are likely a reflection of altered motivational states rather than general hyperactivity" needs to be modified because hyperactivity may be influencing the findings reported

– Regarding the lever pressing extinction data, (1) the authors acknowledge that mice with Ntn1 cKO in GABA VTA neurons have similar extinction rates than WT. They now need to mention that this finding suggests that the increase in the number of lever-pressing during the extinction phase in Ntn1 cKO in GABA mice is most likely associated with an increase in baseline responding rather than with extinction deficits. This same point needs to be highlighted for the Ntn1 cKO in DATIRES::Vgat-Cre mice too. (2) the rate of extinction is not analyzed nor discussed. (3) it is unclear how plotting the extinction bin data for day1 and day 2 and for day 4 and day 5, for suppl Figures5 and 6 respectively, addresses this issue. The authors need to consider removing these data because they do not seem to provide relevant information.

*Reviewer #3 (Recommendations for the authors):*

I appreciate the addition of the bath application of AMPA or NMDA experiments. While these add support to the effect of netrin being postsynaptic, it should be indicated that these experiments do not distinguish between synaptic responses and extrasynaptic responses. Further a change in holding current after bath application of AMPA could be due to reduced synapse number that is associated with a reduced postsynaptic AMPA receptor complement. Further, while they show no change in PPR consistent with no effect on release probability (Pr), their decrease in mEPSC amplitude and frequency could be consistent with a reduction in synapse number (number of release sites N) or a change in number or function of postsynaptic AMPA receptors. To distinguish this, you can measure 1/CV2 which depends on N and Pr but is independent of quantal size. Given that there is no change in Pr, if you see a change in 1/CV2 it might be consistent with a decrease in release sites as opposed to a change in postsynaptic receptor number or function. Alternatively, if there is no change in 1/CV2, you can make a stronger conclusion that the netrin manipulation is altering number or function of postsynaptic AMPA receptors.

Regarding the shift in the E:I balance, while I do not disagree with their model, I was proposing that they could test their model directly by recording mIPSCs at the reversal potential for EPSCs (0 to 10mV) and mEPSCs at the reversal potential for GABA mIPSCs (~-60 to -70mV) and converting the average peak current amplitude into conductance. This way, you can directly measure the amount of inhibitory current and amount of excitatory current the same cell is receiving. Perhaps this experiment is beyond the scope of the current manuscript, but when discussing their model, they could indicate that they could test the validity of this hypothesis in future experiments.

---

## [Author Response]

[Editors’ note: the authors resubmitted a revised version of the paper for consideration. What follows is the authors’ response to the first round of review.]

Reviewer #1 (Recommendations for the authors):The study seeks to investigate whether the guidance cue netrin-1 in the VTA is involved in the regulation of glutamatergic synaptic connectivity of dopamine and GABA neurons and influences behaviors known to be mediated by mesolimbic dopamine systems.The study involves multiple approaches: viral mediated inactivation of NTn1 in VTA GABA or dopamine neurons, using Cre inducible CRISPR/Cas9; in situ hybridization to confirm colocalization of Ntn1 and tyrosine hydroxylase or Ntn1 and SlC31a1; electrophysiological recordings to evaluate changes in neural excitability of dopamine or GABA neurons, a battery of behavioral tests to evaluate whether loss of Ntn1 in either cell type alters sensorimotor gating function and motivated and anxiety-like behaviors.Major strengths of the paper include (i) the novelty of the question addressed and the multidisciplinary approach, (ii) most of the experiments address functional questions, (iii) the careful characterization of functional and behavioral phenotypes for the majority of the experiments, (iv) the generation of a mechanistic model potentially explaining how netrin-1 in the adult VTA, by regulating the excitatory tone of dopamine and GABA neurons, regulates overall dopamine excitability and behavior.Weaknesses of the study are mainly related to the interpretation of their results and the conclusions made: (i) Regarding the results from the in situ hybridization experiment, the authors do not compare their findings with results from a recent single-cell RNA seq study (PMID: 35385745). There is a discrepancy between the findings, particularly in the proportion of GABA neurons expressing NTn1 (as well as in the expression of NTn1 in other cell types, such as astrocytes). The authors do not mention this issue in their results or discussion.

We find that approximately 30% of Vgat-expressing neurons also express Ntn1. Analysis of data from PMID: 35385745 confirms that a subset of neurons in their population designated GABA1, express *Ntn1*. We have also performed snRNA seq in mouse VTA and find that netrin 1 displays partially enriched expression in one of 3 of the major GABA populations that we identified (unpublished data). Differences in the proportion of GABA neurons we identified that express *Ntn1* compared to the data in PMID: 35385745 may reflect differences between rat and mouse, or increased sensitivity of in situ hybridization compared to snRNA seq. We have added discussion of this point to the updated manuscript. The role of *Ntn1* expression in glial cells is interesting, and we have added this point to our revised discussion.

(ii) The authors conclude that their findings demonstrate that netrin-1 regulates the balance of glutamatergic connectivity in the VTA (title, last paragraph introduction, results, and conclusion). Indeed they find that downregulation of Ntn1 in VTA dopamine or GABA cells reduces the frequency and amplitude of their miniature excitatory postsynaptic currents (mEPSCs), without altering miniature inhibitory postsynaptic currents (mIPSCs) or leading to statistically reliable changes in frequency or amplitude in other (non-specified) cell types. They also show that reduced mEPSCs is not mediated by changes in presynaptic release, as revealed by paired-pulse ratio measures. However, these findings are not directly linking electrophysiological changes to alterations in glutamatergic synaptic connectivity.

We apologize for our confusion, but we are not quite clear what the reviewers means when they state the mEPSCs are “not directly linking electrophysiological changes to alterations in glutamatergic synaptic connectivity”. mEPSC and mIPSP frequency and amplitude have been the gold standard in establishing excitatory synaptic connectivity since the original description of quantal synaptic transmission at the motor endplate (Fatt and Katz, 1952 and Del Castillo and Katz, 1954), the introduction of TTX to block action potential firing (Colomo and Erulkar, 1968) and the isolation of miniature inhibitory synaptic currents (Takahashi et al., 1983). A reduction in mEPSC frequency with no change in paired pulse ratio is commonly interpreted as a loss of functional synapses. Additional measures of synaptic connectivity, such as ultrastructural analysis can be performed, but dopamine neurons of the ventral midbrain have long been viewed as generally aspiny neurons (Juraska et al., 1977 J. Comp Neurol, see also Henny et al., 2012 Nat. Neurosci), though there is certainly no clear consensus on this (e.g. Jang et al., 2015 Scientific Reports). To better resolve the observed changes in synaptic connectivity in mEPSCs, we included bath application of AMPA and NMDA in the updated manuscript to quantify the degree of ‘total’ receptors present on surface of these cells with or without *Ntn1* mutagenesis (Figure 3 supplement 3). We believe this addition clarifies the electrophysiological changes that occur with *Ntn1* mutagenesis.

(iii) Whether NTn1 deletion leads to dopamine or GABA neuronal loss remains unknown. This is important considering previous studies linking (or not) changes in the netrin-1 system in VTA neurons and cell loss.

We agree that determining if Ntn1 deletion leads to neuronal loss is an important question. To address this, we have counted TH-positive cells in the VTA of mice with mutagenesis of *Ntn1*, which are more easily assessed by IHC, and determined that loss of *Ntn1* function does not appear to result in the loss of dopamine neurons within the VTA.

(iv) The study shows that downregulation of Ntn1 in dopamine neurons has no significant effect on reward but influences anxiety-like behaviors. In contrast, downregulation of Ntn1 in GABA neurons produces changes in most of the behaviors tested. It is not clear, however, whether increased locomotor activity in mice with Ntn1 deletion in GABA neurons could influence changes in lever pressing in the operant behaviors.

We agree that locomotor effects can confound interpretations in behavioral assays. It is important to note that the level of responding in the FR1 task was not altered in GABA-Ntn1 mutant mice, suggesting that elevated responding in the FR5 and PR assays are not likely a simple reflection of hyperactivity. We have more explicitly addressed this in the revised manuscript.

(v) Regarding the changes in extinction training in the mice with reduced Ntn1 in GABA neurons, it seems that they are lever pressing at a higher rate from the beginning but they extinguish their behavior at a similar rate (similar for the animals with reduced Ntn1 in GABA and in dopamine neurons).

The reviewer makes an interesting point. The extinction burst observed in controls is part of the extinction process that reflects the invigoration of the behavioral response when the outcome does not match expectation at the beginning of the extinction process. We have altered the manuscript to be more specific on what we are referring to regarding the deficits in extinction behavior and have provided additional analysis of the rate of responding on the days of extinction training where differences were observed in the overall numbers of lever presses but not the slopes of the curves.

(vi) the model and hypothesis put forward and tested in the experiments shown in Figure 6 are very interesting. However, the results obtained do not justify the conclusion that Loss of Ntn1 function in both cell types simultaneously largely rescues the consequences induced by GABA- only Ntn1 deletion.

We apologize for the lack of clarity. We have altered the discussion to more explicitly provide an explanation of how mutagenesis in both cell types can result in a diminution of the observed phenotype in the GABA-Ntn1 knockout mice. In the model proposed, reduced excitatory input onto GABA neurons results in a reduced inhibition of dopamine neurons causing the observed phenotypes. For reference, we have previously shown that blocking all synaptic transmission from GABA neurons in the VTA (Gore et al., 2017 *ELife*), or blocking selectively GABA release from VTA GABA neurons (Soden et al., 2020, *Nat. Neurosci.*) results in hyperactivity and increased operant responding, though to a much greater degree than the effects observed here. We, and others have also shown that loss of glutamate signaling in dopamine neurons has only a modest behavioral effect (Zweifel et al., 2009 PNAS, and Hutchinson et al., 2017 Mol Pscyh.), so it is not surprising that reducing mEPSCs onto VTA dopamine neurons has little effect in this context. The partial rescue of the GABA-Ntn1 knockout experiment suggests that reducing glutamatergic input onto dopamine neurons can abrogate effects associated with reduced inhibitory tone, thus restoring the excitatory and inhibitory balance. In the revised discussion we outline a detailed mechanism that we propose explains this observation.

To be able to link the electrophysiological changes to glutamatergic synaptic connectivity, other experiments are required, including assessing structural changes (e.g. PMID: 24174661) in dopamine and GABA neurons as well as the proportion of AMPA/NMDA receptors and AMPA/NMDA ratios. In this regard, there is a previous study relating the netrin-1 guidance cue system in adult VTA synaptic plasticity (PMID: 20345916).

An initial address of this point was provided above. Assessing AMAPA/NMDA ratio is another measure of synaptic strength; however, this assumes that NMDA receptor levels remain unchanged, which we know from previous studies is not always true in the VTA (Mameli, M., Bellone, C., Brown, M. T. & Lüscher, C 2011 Nature). To address this, we recorded both AMPA and NMDA-evoked current in control and *Ntn1* mutant dopamine neurons. We have also made sure that we reference the above highlighted paper which describes the impact of Dcc heterozygous loss of function on amphetamine sensitization.

Whether NTn1 deletion leads to dopamine or GABA neuronal loss could be addressed using stereology.

We have included a count of the number of virally labelled dopamine neurons to address this point and show that *Ntn1* deletion did not result in a loss of dopamine neurons.

To be able to conclude that the loss of Ntn1 function in both cell types simultaneously largely rescues the consequences induced by GABA- only Ntn1 deletion, the electrophysiological properties of dopamine (and also GABA) neurons could be assessed. This experiment will also test more directly the model the authors are proposing.

This experiment is more difficult to address than the reviewer may appreciate. Both dopamine and GABA neurons express Cre, so fluorescent isolation of these cells will not be possible. Ih currents can be used, but not all dopamine neurons are Ih positive. This is particularly problematic as the number of excitatory synapses onto GABA neurons is higher than onto dopamine neurons, thus there will be a large spread in the distribution of responses making statistical analysis difficult to achieve. To address this, we have been more explicit in our discussion of what we hypothesize the double mutagenesis is achieving and why this is important for considerations of the cell-type and circuit-specific impacts of *Ntn1* loss of function are and how they are consistent with what is known about the organization of the VTA cell types.

In the discussion, evidence showing the role of glutamatergic inputs of VTA dopamine neurons on behavior needs to be revised more carefully (e.g. PMID 25388237, PMID 26631475, and PMID 30699344).

We have expanded this area of the discussion to address this point.

Reviewer #2 (Recommendations for the authors):This manuscript by Cline et al. sought to define the role of the axonal guidance cue netrin-1 in synaptic signaling in the ventral tegmental area (VTA). The authors used CRISPR-Cas9 mutagenesis to reduce netrin-1 expression/function in the two predominant neuronal types in the VTA, dopaminergic and GABAergic neurons. This work builds on previous work from this group showing a role for the axon guidance receptor ROBO2 in inhibitory connectivity in adult VTA. Netrin-1 is examined here because of its persistent expression in the VTA into adulthood, despite its established role as a developmental protein. A strong combination of techniques is used, including selective knockdown of the netrin-1 gene in multiple neuron types in the VTA, patch clamp electrophysiology, and several behavioral assays. The results clearly indicate that knockdown of netrin-1 specifically in either GABA or dopamine neurons reduces miniature glutamatergic synaptic currents specifically in that cell type. Interestingly, inhibitory input was not significantly affected. This is consistent with previous reports of effects on excitatory synapses in the hippocampus. Robust behavioral consequences were only reported in the GABA neuron knockdown and were not evident when netrin-1 was knocked down in both cell types. The authors conclude that netrin-1 is important for maintaining the balance of excitatory input onto the two main neuronal subtypes in the VTA. This conclusion is largely supported by the results from the experiments, which were performed rigorously. The manuscript itself was easy to follow and well written, save for some minor omissions.Operant responding for food was used as one of the dependent measures in Figures 4 and 5, with differences observed in the GABA neuron knockdown, however, one omission of the study was that body weights of the mice before and especially after treatment were not reported. It may be that viral knockdown of netrin-1 in one cell type has effects for instance on satiety, producing effects on behavior along with energy balance. The addition of body weight data would help round out the data set and, if different, might help with interpretation.

This is an excellent point. All mice were food restricted to 85% of their body weight, and the methods have been changed to reflect that omission. Initial bodyweights at the start of calorie restriction have also been included in the supplemental data.

Details were also not provided for the food restriction procedure that was used during operant conditioning for food pellet responding, which could have further interacted with the netrin-1 manipulation in the mouse lines to affect physiology, behavior, or both.

We apologize for the oversight and have corrected the methods to reflect the calorie restriction used.

As discussed above, most of this study is strong, with interpretations supported by largely convincing results. However, the manuscript could be improved with additions and clarifications.While both cell types showed increased mEPSC frequency and amplitude after netrin-1 knockdown, only the GABA neuron knockdowns showed robust behavioral effects, including increased locomotion (day and night), and increased operant responding for food, and decreased acoustic startle and pre-pulse inhibition. As some of the behavioral tasks involve responding for food pellets, interpretation of the results would benefit from reporting the weights of the mice before and after treatment. Additionally, details about the food restriction procedure that was used for operant conditioning should be provided. Several reports have identified the effects of the feeding state on dopamine neuron excitability and synaptic input to the area. Was the food restriction controlled to a percent loss of body weight, applied acutely or chronically, done throughout the operant study or just during FR1, etc.? Did the mice always eat the pellets when performing the operant task? It's interesting that responding in all of the dopamine mice stayed at the same values when they switched from FR1 to FR5, but in the GABA mice, the control animals fell while the netrin-1 knockdowns stayed the same. Full disclosure of these details would help the reader interpret small observations in the data and later assist in reproducing the results, should they wish to do so.

We apologize for the oversight and have corrected the methods to reflect the food restriction used. We have also included the body weights of the animals prior to calorie restriction.

The evidence for the anxiety behavioral phenotype in the dopamine neuron-specific netrin-1 knockdown is pretty thin, as a time-in-center measure in an open field could be affected by other factors, and other supporting data for instance from elevated mazes were not used. Admittedly, anxiety can be difficult to show in mice. However, Reference 20 which was used to support the "proposed role of dopamine in the modulation of anxiety-related behavior" was a dead-end non-citation. If the authors wish to keep this part of their interpretation they should bolster the explanation of the link between dopamine and anxiety with further evidence (experimental or literature). Also, details about the meaning of "time in center" and "time on edge" should be provided in the Methods.

We apologize for the broken citation. The citation should have read ” Zarrindast MR, Khakpai F. The Modulatory Role of Dopamine in Anxiety-like Behavior. *Arch Iran Med*. 2015;18(9):591-603.” The reference has been corrected.

We have also provided additional information re: time in center and time on edge in the methods to add clarification.

The extinction result in the GABA mice is presented as "a significant delay in the rate of extinction following reinstatement of FR1." This is hard to interpret because responding immediately before extinction is not given and the data are presented only as raw numbers instead of percent of baseline. If the netrin-1 knockdowns were responding more at FR1 when they were returned to that condition, the shape of the curve would actually indicate no difference in extinction. The same could be true in Figure 6 with the double cell knockdowns. Depending on what the data look like, these graphs may be more accurately presented as normalized numbers. This is a minor issue in the scheme of this paper but it should nonetheless be clarified.

We have provided the data on the reinstatement training and have performed additional analysis of the rates of extinction for each of the mice within a given session to more accurately address this concern.

The schematics in Figure 6A show that the effects on the GABA neurons proceed through the dopamine neurons. While this is entirely plausible, the authors never actually show this experimentally, for instance by locally manipulating GABA input in virus-injected mice. It may be at least as likely that the important interactions occur in the nucleus accumbens, or some other area to which both cell types project. As a minimum, the authors should point out this caveat in the Discussion, or point out any other possibility that could also explain the somewhat surprising data in Figure 6.

We agree that the projection of a subset of GABA neurons to other brain regions may play an important role in the observed phenotypes. We have clarified this in the discussion.

There's some confusion about the Ntn1 co-localization in Figure 1. The language in the caption and Figure 1 itself seem clear. However, the text on page 3 seems to contradict the language in the figure caption. The way it is worded, instead of 64, 30, and 6% shouldn't these numbers be 72, 18, and 10%? Please clarify with the correct information, or point out where the confusion lies.

We apologize for the confusion and have clarified this in our revised manuscript.

In the abstract, "simultaneously" is placed confusingly in the sentence about rescuing the GABA phenotype. This would be more clear if the sentence started "Simultaneous loss… in both cell types."

We agree and have made this change.

Figure 1D has two (identical) scale bars.

We have corrected this inadvertent duplication.

Reviewer #3 (Recommendations for the authors):In this study, they used genetic strategies to decrease Netrin-1 expression in either dopamine or GABA neurons of the VTA. Reduction of Netrin-1 expression in VTA dopamine neurons decreased excitatory, but not inhibitory postsynaptic currents onto TH^+^ or GABA VTA neurons. While the loss of netrin-1 in dopamine neurons did not significantly influence behaviour, loss of netrin-1 in GABAergic neurons increased locomotor activity, effort for rewards, extinction delay, and decreased prepulse inhibition. Finally, effects on locomotor activity, reward seeking, and prepulse inhibition observed in GABA-netrin mice were not present when there was the loss of netrin-1 in both dopamine and GABA neurons, suggesting that loss of netrin-1 in both could restore the behavioural effects of loss of netrin-1 in GABA neurons.Major strengths:This is a nicely written, clearly illustrated study describing the loss of netrin-1 in VTA dopamine neurons and GABA neurons. The authors use an elegant genetic methodology to knock down netrin-1 in select populations of neurons within the VTA. They use a battery of behavioural assays to examine the effects of this knockdown.Major weaknesses:While this study provides a potential physiological mechanism underlying the changed behavioural effects, it doesn't connect these changes to the behaviour or identify the mechanism by which adult netrin-1 influences these changes in synaptic transmission. While netrin-1 is involved in synapse formation during development, it is not clear how netrin-1 is influencing synapses in adulthood. For example, is it necessary for the stabilization of synapses?

We agree with the reviewer and have received a similar concern from reviewer 1. A response to this is provided above and we will summarize our experimental approach below.

Author suggestions:The authors propose that loss of netrin-1 in dopamine neurons leads to enhanced excitation and decreased inhibition, whereas loss in GABA neurons would lead to enhanced inhibition and decreased excitation, and the E:I ratio would be balanced when netrin is lost from both cell population. However, they do not test this assertion by measuring excitatory:inhibitory ratio in each model. This would support their hypotheses in figure 6.

We thank the reviewer for their comment. We conclude that loss of *Ntn1* function in either dopamine neurons or GABA neurons results in a loss of excitatory synaptic connectivity with no loss in inhibitory synaptic connectivity. Loss of excitatory input with no change in inhibitory input will bias towards inhibition in the E:I balance. When this occurs in GABA neurons, because local GABA potently inhibits dopamine neurons, the loss of excitation onto GABA neurons biases towards greater inhibition of these cells which disinhibits the dopamine neurons causing a hyperdopaminergic phenotype. We have provided a detailed discussion of the potential mechanism by which loss of *Ntn1* in both cell types would restore the E:I balance.

Some of the discussion focuses on the role of netrin-1 in development. However, the manipulations done in this paper were to remove netrin-1 in adulthood after axon migration and synapse formation occur. They do not discuss what the 'adult function' of netrin-1 is. While it seems to play a role in excitatory synaptic transmission, given its effects on mEPSCs, the authors did not provide sufficient information to conclude if this was a reduction in the number of synapses, a silencing of synapses, or a decrease in release probability with compensatory postsynaptic changes. Experiments addressing how adult netrin-1 signaling in the VTA specifically influences synaptic transmission onto dopamine or GABA neurons of the VTA may highlight how netrin-1 might be contributing to associated behavioral changes.

As mentioned above, we have included recordings of AMPA and NMDA receptor mediated currents (via bath application of the specific agonists) in these cells to quantify the levels of these receptors. Our paired-pulse ratio experiments indicate that presynaptic release probability is not altered with the loss of *Ntn1* pointing to a postsynaptic effect.

[Editors’ note: what follows is the authors’ response to the second round of review.]

Essential revisions:We have a few remaining concerns that can be addressed with an additional analysis of existing data (see R3 point #1) and edits to the text to add clarity, change language, temper conclusions, discuss alternatives, etc.Please provide a point x point response to each reviewer point along with your revision.Reviewer #1 (Recommendations for the authors):This resubmitted manuscript is substantially improved from the previous version. Additions and clarification to methods and interpretations have largely addressed my previous concerns, which were minor. Additional discussion has bolstered the notion that netrin-1 is impotant for maintaining the balance of excitatory and inhibitory inputs to dopamine neurons. The addition of maze data would have bolstered conclusions about the anxiety phenotype, but "time on edge" and locomotor data was added and this was always a minor point that did not detract substantially from the rest of the manuscript.One item does need to be clarified. The experiment represented in Figure 3 Supplement 3 was performed to address a concern of Reviewer 1, however the text describing this is very confusing. The Results describe these data as evoked EPSCs following bath perfusion of AMPA or NMDA. The data themselves seem to be showing changes in holding current in a voltage clamp experiment, although the axis labels and the lack of sample traces leave this in doubt. If it is holding current, more details should to be provided in the figure (including better axis labels and a description of how long the drug went on and the holding voltage, both of which are currently only in the Methods). If instead it is evoked EPSCs in the presence of bath perfused agonists, then other details also need to be provided to make this result make sense (such as sample traces and an explanation of what they were after).

We apologize for the confusion and lack of clarity regarding these experiments. These data were from a voltage clamp experiment in which agonist was bath applied for 30 seconds, and holding current was recorded. The figure legends and axis have been amended to reduce confusion, and the experiment has been better clarified in the Results section.

The addition of body weight data and food restriction information is appreciated. However in the Response to Reviewers the level of restriction was listed as 85% of initial body weight, whereas in the manuscript this is given as 80%. Please make sure the correct number appears in the manuscript.

The addition of body weight data and food restriction information is appreciated.

However in the Response to Reviewers the level of restriction was listed as 85% of initial body weight, whereas in the manuscript this is given as 80%. Please make sure the correct number appears in the manuscript.

Reviewer #2 (Recommendations for the authors):The revised version improved the interpretation of the results, and I am pleased that the authors replied to most of my comments on the previous version. They added needed citations, clarification on their manipulation, and additional experiments to understand the changes in synaptic connectivity better and clarify the electrophysiological changes that occur with Ntn1 mutagenesis, as well as their proposed model.The following issues remain unresolved:– Since the study does not assess alterations in neuronal structure and connectivity, we suggest the word "connectivity" to be dropped or modified.

We thank the reviewer for their comment. Although we disagree that the measurement of miniature inhibitory and excitatory postsynaptic currents is not a direct measure of synaptic connectivity, we have changed the title to “Netrin-1 regulates the balance of synaptic glutamate signaling in the adult ventral tegmental area”.

– It appears that stereology was not performed to calculate the number of neurons expressing netrin-1 in TH^+^ or GABA+ cells in the mice with conditional netrin-1 KO. The statement "Total Th-positive cells were recorded for all images and averaged across all slices for each mouse to give a total number of Th-positive cells" needs to be revised, because total number of cells can only be assessed using stereological analysis. It remains unknown whether NTn1 deletion leads to dopamine or GABA neuronal loss. This is an important issue that needs to be acknowledged in the manuscript. Please seehttps://www.embopress.org/doi/full/10.15252/embj.2020105537

We thank the reviewer for this comment. We have addressed this with the following statement in the Results section: “Following *Ntn1* deletion the average number of TH^+^ cells per slice was not statistically different in DAT-Cre *Ntn1* cKO mice compared to controls (control: 178.2±12.68 and *Ntn1* cKO 173.3±10.75). Although this result is consistent with *Ntn1* inactivation not compromising cell viability, without a complete stereological analysis of every neuron within the VTA, we cannot definitively conclude that some cell loss did not occur.”

– Regarding the increase in locomotor activity observed after the downregulation of Ntn1 in GABA neurons, the authors argue that the level of responding in the FR1 task is not altered in GABA-Ntn1 mutant mice, suggesting that elevated responding in the FR5 and PR assays is not likely a reflection of hyperactivity. Yet, increasing the response rate in the FR1 protocol is no possible. Once the animals respond to the reward in the FR1, the lever retracts, and no more lever-pressing is possible until a reward is collected. The statement added by the authors "however, we did not observe differences in operant responding during FR1 training or reinstatement indicating that the observed effects are likely a reflection of altered motivational states rather than general hyperactivity" needs to be modified because hyperactivity may be influencing the findings reported

We appreciate this suggestion and have modified the phrasing in the manuscript in response. The phrasing has been changed to: While these data likely reflect altered motivational state with loss of Ntn1, it is also possible that the hyperactivity observed in Vgat-Cre *Ntn1* cKO mice contributes to the elevated lever press rates during FR5, PR, and extinction.

– Regarding the lever pressing extinction data, (1) the authors acknowledge that mice with Ntn1 cKO in GABA VTA neurons have similar extinction rates than WT. They now need to mention that this finding suggests that the increase in the number of lever-pressing during the extinction phase in Ntn1 cKO in GABA mice is most likely associated with an increase in baseline responding rather than with extinction deficits. This same point needs to be highlighted for the Ntn1 cKO in DATIRES::Vgat-Cre mice too. (2) the rate of extinction is not analyzed nor discussed. (3) it is unclear how plotting the extinction bin data for day1 and day 2 and for day 4 and day 5, for suppl Figures5 and 6 respectively, addresses this issue. The authors need to consider removing these data because they do not seem to provide relevant information.

We thank the reviewer for this comment and we believe that this is addressed in the response above. With regard to the inclusion of the additional extinction analysis previously requested by the reviewer, we believe that this data provides additional information for the reader to better understand the extinction responding in control and *Ntn1* mutant mice.

Reviewer #3 (Recommendations for the authors):I appreciate the addition of the bath application of AMPA or NMDA experiments. While these add support to the effect of netrin being postsynaptic, it should be indicated that these experiments do not distinguish between synaptic responses and extrasynaptic responses. Further a change in holding current after bath application of AMPA could be due to reduced synapse number that is associated with a reduced postsynaptic AMPA receptor complement. Further, while they show no change in PPR consistent with no effect on release probability (Pr), their decrease in mEPSC amplitude and frequency could be consistent with a reduction in synapse number (number of release sites N) or a change in number or function of postsynaptic AMPA receptors. To distinguish this, you can measure 1/CV2 which depends on N and Pr but is independent of quantal size. Given that there is no change in Pr, if you see a change in 1/CV2 it might be consistent with a decrease in release sites as opposed to a change in postsynaptic receptor number or function. Alternatively, if there is no change in 1/CV2, you can make a stronger conclusion that the netrin manipulation is altering number or function of postsynaptic AMPA receptors.

This is an excellent suggestion. We calculated 1/CV^2^ for mEPSCs based on this suggestion and have presented that data in the results and as figure 3 supplemental 3.

Regarding the shift in the E:I balance, while I do not disagree with their model, I was proposing that they could test their model directly by recording mIPSCs at the reversal potential for EPSCs (0 to 10mV) and mEPSCs at the reversal potential for GABA mIPSCs (~-60 to -70mV) and converting the average peak current amplitude into conductance. This way, you can directly measure the amount of inhibitory current and amount of excitatory current the same cell is receiving. Perhaps this experiment is beyond the scope of the current manuscript, but when discussing their model, they could indicate that they could test the validity of this hypothesis in future experiments.

We appreciated the suggestion and do agree that this experiment is beyond the scope of the current manuscript, but we have included this as a suggestion for future experiments in the discussion of this paper.